# Advanced Hydrogels Combined with Silver and Gold Nanoparticles against Antimicrobial Resistance

**DOI:** 10.3390/antibiotics12010104

**Published:** 2023-01-06

**Authors:** Yolice Patricia Moreno Ruiz, Luís André de Almeida Campos, Maria Andressa Alves Agreles, André Galembeck, Isabella Macário Ferro Cavalcanti

**Affiliations:** 1Laboratory of Microbiology and Immunology, Academic Center of Vitória (CAV), Federal University of Pernambuco (UFPE), Vitória de Santo Antão 55608-680, Pernambuco, Brazil; 2Department of Fundamental Chemistry, Federal University of Pernambuco (UFPE), Av. Jorn. Aníbal Fernandes, Cidade Universitária, Recife 50740-560, Pernambuco, Brazil; 3Institute Keizo Asami (iLIKA), Federal University of Pernambuco (UFPE), Av. Prof. Moraes Rego, 1235, Cidade Universitária, Recife 50670-901, Pernambuco, Brazil

**Keywords:** antibiotic resistance, hydrogel, bacterial, antibiofilm, nanotechnology

## Abstract

The development of multidrug-resistant (MDR) microorganisms has increased dramatically in the last decade as a natural consequence of the misuse and overuse of antimicrobials. The World Health Organization (WHO) recognizes that this is one of the top ten global public health threats facing humanity today, demanding urgent multisectoral action. The UK government foresees that bacterial antimicrobial resistance (AMR) could kill 10 million people per year by 2050 worldwide. In this sense, metallic nanoparticles (NPs) have emerged as promising alternatives due to their outstanding antibacterial and antibiofilm properties. The efficient delivery of the NPs is also a matter of concern, and recent studies have demonstrated that hydrogels present an excellent ability to perform this task. The porous hydrogel structure with a high-water retention capability is a convenient host for the incorporation of the metallic nanoparticles, providing an efficient path to deliver the NPs properly reducing bacterial infections caused by MDR pathogenic microorganisms. This article reviews the most recent investigations on the characteristics, applications, advantages, and limitations of hydrogels combined with metallic NPs for treating MDR bacteria. The mechanisms of action and the antibiofilm activity of the NPs incorporated into hydrogels are also described. Finally, this contribution intends to fill some gaps in nanomedicine and serve as a guide for the development of advanced medical products.

## 1. Introduction

Antimicrobial drug resistance is considered one of the greatest threats to global public health. Multidrug resistance (MDR) is of particular concern, especially among ‘ESKAPE’ organisms: *Enterococcus faecium*, *Staphylococcus aureus*, *Klebsiella pneumoniae*, *Acinetobacter baumannii*, *Pseudomonas aeruginosa*, and *Enterobacter* spp., as they are responsible for many nosocomial severe infections [1]. Multi-resistant bacteria such as *P. aeruginosa*, *A. baumannii*, and *Enterobacteriaceae* have been declared priority one pathogens in a “List of Priority Pathogens for Research and Development of New Antibiotics” [2]. According to the World Health Organization (WHO), antibiotic resistance is one of the biggest public health problems, and around 79% of bacteria have developed resistance to one or more antibiotics. Approximately 700,000 people worldwide die from drug-resistant bacterial infections; by 2050, this number is estimated to reach 10 million [3,4]. In the US only, the healthcare cost related to antibiotic resistance is approximately $20 billion per year [4,5,6].

Antibiotic resistance comes from changes in the structure of bacteria due to changes in the genetic material or through the acquisition of genetic material from external sources, such as viruses, other bacteria, and the environment [7]. Moreover, it is known that about 40 to 80% of bacteria can form biofilms [8]. Biofilms are sets of sessile microorganisms attached to a substrate, or to each other, enclosed in a self-produced polymeric matrix. When embedded in the matrix, bacterial cells exhibit an altered rate of growth and gene transcription. Additionally, they produce specific proteins that help them to become more resistant, and they present high gene exchange, raising the recombination rate among the strains [9,10]. According to the National Institute of Health, biofilms are responsible for up to 80% of all microbial infections in humans, including cases of endocarditis, cystic fibrosis, non-healing chronic wounds, meningitis, kidney infections, and infections related to implantable devices such as urinary catheters, prosthetic joints, and heart valves [11].

Both innate and acquired host immune responses are activated during a biofilm infection. However, neither of these immune responses can eradicate the pathogen in the biofilm due to the polymeric matrix, which acts as a structural barrier. In addition, sessile bacteria are less responsive to traditional antibiotic therapy because they are 500 to 5000 times more resistant to drugs than planktonic cells. Thus, new strategies to inhibit biofilm formation and to eradicate already formed ones, are mandatory [10,12].

Amongst the pathogens with MDR to antibiotics, these bacteria can be highlighted: *S. aureus*, resistant to methicillin and vancomycin (MRSA and VRSA, respectively); *E. faecium*, resistant to vancomycin or fluoroquinolone; *E. coli*, resistant to polymyxin; and *Acinetobacter* spp., which is resistant to aminoglycosides [13,14]. Between 2018 and 2019, the United States Food and Drug Administration (US FDA) approved nine new antibiotics from 107 molecules, to fight against MDR bacteria [15].

Numerous studies have aimed to understand the phenotypic and genotypic evolution of antibiotic resistance [16]. Although some promising agents have been explored [17,18], there is an urgent need for new active antibiotic molecules, but usually, new antibiotics take a long time to be developed [4]. Thus, to reduce the problem of antibacterial resistance in a short period, will be challenging.

Before the discovery of penicillin, certain metals, oxides, or metallic salts were used to treat bacterial and fungal infections [19], but their use have declined. In recent years, inorganic materials, especially nanostructured systems, have proven to be effective against pathogenic microorganisms [20,21]. Nanoparticles of metals and metal oxides, such as silver (Ag), gold (Au), MgO, ZnO, and TiO_2_, with antimicrobial activity, have been proposed as antibiotics (Table 1) [4].

Silver nanoparticles (Ag NPs) and gold nanoparticles (Au NPs) have been proposed as a new class of antibiotics. These NPs have shown a broad antibacterial activity against *E. coli* [22,23,24], *S. aureus* [25,26], *P. aeruginosa* [27,28], *Proteus vulgaris*, *S. aureus*, *Proteus mirabilis* [13], *Enterobacter cloacae* [29], and *Staphylococcus epidermidis* [28]. These metallic NPs are able to inhibit the growth of bacteria by inhibiting the formation of bacterial biofilms and/or destroying pathogenic microorganisms [30,31].

For in vivo application, the colloid of NPs requires a platform that acts as a carrier, providing stability to the NPs, regulating their controlled release at the local site of the bacterial infection. Among the supportive materials for metallic NPs, hydrogels are the most commonly investigated in nanomedicine. Hydrogels are tridimensional structures that can be functionalized due to the presence of distinct functional groups inside the network. Recently several antimicrobial agents have been incorporated into hydrogels such as antibiotics [32], nanoparticles [22,23,24], bacteriophages, antibacterial peptides [33], biological extracts [34,35], and antimicrobial enzymes [36]. Some of the prerequisites of hydrogels for health applications are: non-toxicity, sustainability, environmentally friendly [13], flexibility, elasticity [37], biocompatibility [38], immunogenic [22], biodegradability [39], resistance to severe conditions [40], good extensibility [41], and the ability to stimulate nutrients and metabolic exchange [22]. Hydrogels are capable of improving cellular internalization [29], absorbing wound exudates [42], expediting skin healing, stimulating the collagen proliferation [40], and exhibiting antibiofilm activity [23,37,39,43,44]. They may also be applied in medical devices such as venous or urinary catheters, artificial voice prosthesis, and prosthetic heart valves [45].

**Table 1 antibiotics-12-00104-t001:** Examples of metallic NPs used against resistant bacteria and their mechanism of bactericidal action.

NPs	Bacteria	Mechanism of Action	Ref.
Ag	*E. coli*, *B. subtilis*, and *S. aureus*	Ag^+^ ion liberation; Cell membrane destruction and electron transport; Bacterial DNA damage	[20,46]
Au	*P. aeruginosa* and *E. coli*	Interaction with Au NPs Mg^2+^ or Ca^2+^ ion sequestration to damage the cell membrane Competition for the virus binding to the cell	[47,48,49,50]
ZnO	*E. coli*, *S. aureus*, and *Botrytis cinerea*	Intracellular NP accumulation Damage to the cell membrane H_2_O_2_ production Zn^2+^ ion liberation	[51,52,53]
TiO_2_	*E. coli* and *Bacillus megaterium*	Production of active oxygen species Cellular membrane destruction Generation of electron-hole pair by visible light excitation with low recombination rate	[54,55]
Cu	*E. coli* and *Bacillus subtilis*	Cu^2+^ ion liberation Cellular membrane damage DNA alteration	[56]
MgO	*E. coli*, *S. aureus*, *Bacillus subtilis*, and *Bacillus megaterium*	Cellular membrane damage Alkalinization by MgO Hydration Active oxygen liberation	[57,58]

Ag NPs and Au NPs are loaded into polymer-based hydrogels with a porous structure, such as alginate [59], chitosan [27,60], gelatin [61], konjac glucomannan [23], hydroxypropyl methylcellulose [24], carboxymethyl cellulose [13], carboxymethyl chitosan [62], polyvinyl alcohol [37], carbopol [63,64], gelatin methacrylate [65], polyacrylamide [40], polyethyleneimine [66], and polyvinylpyrrolidone [40,63,67,68,69]. Occasionally, hydrogels exhibit poor mechanical properties, and other agents are required as additives in the manufacturing process, such as tannin acid [60], graphene [66,70,71,72], aluminosilicate nanotubes (NTs) [65], and metal-organic frameworks (MOFs) [69].

In this review, several significant aspects are presented, such as (i) biocompatible natural and synthetic polymers; (ii) synthesis strategies to produce antibacterial hydrogels; (iii) the physical, chemical, and biological properties of the hydrogels and the NPs; (iv) the synergism between the hydrogels and NPs characteristics; (v) NPs aspects that stand out in the antibacterial or antibiofilm efficiency; (vi) mechanisms of antibacterial action, of action for inhibiting biofilm, and for biofilm eradication.

## 2. Silver Nanoparticles (Ag NPs)

Ag NPs can damage the extracellular membrane of bacteria and their intracellular components, exhibiting a broad-spectrum antimicrobial effect [4]. Many Ag NP synthesis strategies have been developed to allow specific Ag NP surface properties, which, in turn, strongly depend on the characteristics of the reducing agent and the type of stabilizer used during their synthesis [4,73,74,75,76,77,78,79].

According to Sondi et al. [80], Ag NPs can cause harm to *E. coli* by forming pits in the cell wall, which could increase its permeability and affect the membrane vesicles. Such damage has also been observed in other bacteria, such as *Scrub typhus*, *P. aeruginosa*, and *Vibrio cholerae* [81], which is attributed to the ability of Ag NPs to interact with some of its components, such as lipopolysaccharides (LPSs), and phosphatidylethanolamines (PEs) [82].

Mirzajani et al. [83] suggested that the ability of Ag NPs to harm the bacterial cell wall may result from their interaction with the peptidoglycan layer, since Ag NPs attack the β-1-4 bonds of N-acetylglucosamine and N-acetylmuramic acid of the glycan chain in the cell membrane of *S. aureus*. Additionally, Ag NPs may produce free radicals, such as reactive oxygen species (ROS), inside and outside the bacteria [84,85]. Elevated ROS levels are known to damage cell DNA, proteins, and enzymes, which could, in turn, interfere with the normal metabolism of bacteria [86]. It was found that Ag^+^ ions released from Ag NPs can damage bacterial membrane function. In particular, the differences in Ag^+^ concentration can induce a difference between the pH and the electrical potential inside and outside the membrane vesicles in *Vibro cholerae*, leading to the failure of membrane respiration and H^+^ leakage [87].

The effect of Ag NPs on the bacterial membrane is related to their physicochemical properties, such as size, shape, surface area, surface charge, oxidation state, and surface chemistry. It has been reported that Ag NPs with small size and colloidal stability are preferred rather than those susceptible to aggregation [4,13,22,66,73,88]. The size of NPs is one of the most critical aspects determining their interaction with cells. Actually, Ag NP interaction is size-dependent [81,89]. Several works have shown that Ag NPs with a diameter of 3–10 nm, are the most effective in killing bacteria due to their preferential direct interaction with the bacterial membrane [52] and how fast the bacterial killing took place after their interaction [89].

The shape of Ag NPs can directly influence the available contact area needed to facilitate interactions of Ag NPs with the bacterial membrane. A comparative study using polyvinylpyrrolidone (PVP)-coated Ag NPs with different shapes suggested a strong correlation between the shape of the Ag NPs and their bactericidal properties. For example, Ag nanoplates (two-dimensional structure, 2D) showed the highest antimicrobial activity against *S. aureus* and *E. coli*, when compared to Ag nanorods (one-dimensional structure, 1D) and spherical Ag NPs (zero-dimensional structure, 0D). Sadeghi et al. [90] showed that Ag nanoplates exhibited the largest surface area, providing the most significant contact area to interact with the bacterial cell wall.

The Ag NP surface charge is also important. It was observed that positively charged Ag NPs using capping agents such as poly(amide amine) dendrimers (PAMAM) [91], poly(ethyleneimine) (PEI) [92], poly(ethylene glycol) (PEG), and polyvinylpyrrolidone (PVP) [67] facilitated the interaction between the particles and the negatively charged bacterial membrane [91]. Ag NPs with a negative surface charge have shown lower antimicrobial activity [93] due to the strong repulsion between the particles and the bacterial wall. This limits the interaction between Ag NPs and bacteria and considerably weakens their antimicrobial effect.

Ag NPs have been also combined with antibiotics (ampicillin, amoxicillin, chloramphenicol, erythromycin, among others) [94] by chelation of the active groups. The combination of Ag NPs with other materials, such as polycationic chitosan, has shown promising results by facilitating the attachment of Ag NPs to the negatively charged bacterial wall [95]. Mishra et al. [96] developed a multifunctional system of Ag NPs embedded in the chitosan-polyethylene glycol (CS-PEG) hydrogel. This implantable device inhibited biofilm formation and the released the drug payload at the same time. Chen et al. [97] prepared a chitosan sponge containing Ag NPs and used it as a tissue for wound healing. Both in vitro and in vivo composite tests showed excellent antibacterial activity against drug-resistant pathogenic bacteria.

Recently, researchers discovered that Ag nanoclusters (NCs) are effective for this type of application [98,99]. NCs are NPs whose sizes are smaller than 2 nm and contain “countable” Ag atoms as a nucleus protected by organic ligands [4]. Ag NCs have shown promising results for biomedical applications, such as bioimaging, biosensing, and antimicrobial agents [100,101]. These NPs have also been used to functionalize natural cellulose nanofibers [102], silk fibers [103], textiles [104], and natural or synthetic polymer-based hydrogels to exhibit antimicrobial activity. Although there are many studies of antimicrobial Ag NPs embedded into hydrogels as platforms for delivering metallic nanostructures as alternative to standard drugs; their mechanism of action has not been entirely elucidated. Nevertheless, all the above examples demonstrate this as a promising strategy in preventing and eradicating infections [7,105].

### 2.1. Antibacterial Activity of Ag NPs Loaded into Hydrogels

Ag NPs incorporated into hydrogels have shown antibacterial properties and the ability to control infections [37]. The NPs are incorporated into a hydrogel with a porous structure by in situ polymer synthesis or by adding the NP colloid to the polymer. Additionally, microwave radiation is another approach to produce NPs within hydrogels. The polymer-based hydrogels help to control the morphology and size of the nanostructures and participate as a stabilizing medium for nucleation sites to produce silver seeds [13,66]. Biocompatible polymers, such as chitosan [27], konjac glucomannan [23], carboxymethyl cellulose [13], carboxymethyl chitosan [62], polyvinyl alcohol [37], carbopol-934 [63], graphene [41,70,71,72], gelatin methacrylate [65], polyacrylamide [40], polyethyleneimine [66], and polyvinylpyrrolidone [40,63,67,68,69], have been used to fabricate antibacterial and antibiofilm materials.

These polymeric biomaterials have helped to treat and prevent infections caused by pathogenic bacteria and are capable of improving the healing and regeneration of the skin. For example, Ag/chitosan/hydrogel has been shown to help the healing process, reduced inflammation at skin wounds, and accelerated the re-epithelization rate to treat post-operative infection [22].

Chitosan (CS) is derived from chitin, the second most abundant biopolymer in nature, after cellulose. It has been used in the synthesis of hydrogels due to its biodegradability, biocompatibility, and antibacterial activity [22]. Chemical crosslinking, the addition of nanofillers, blending with other polymers, and using alkali–urea solutions, are some of the methods used to improve chitosan processability. To improve the water solubility of chitosan, quaternization method has been used, in which a quaternary ammonium moiety was introduced into the chitosan structure by chemical reactions, thus producing quaternate chitosan. Some studies have reported the use of chitosan as a matrix to incorporate Ag nanoparticles. Ag NPs were also synthesized in situ within oxidized dextran (ODex), adipic dihydrazide-grafted hyaluronic acid (HA-ADH), and quaternized chitosan (HACC), resulting in the Ag–ODex/HA-ADH/HACC hydrogel [27]. The Ag NPs had a particle distribution size of around 50–190 nm. The hydrogel displayed antibacterial properties against *E. coli* ATCC 8739, *S. aureus* ATCC 14458, and *P. aeruginosa* CMCCB10104, and the inhibition zones were 24, 24, and 27 mm, respectively. These results were associated with the hydrogel’s positive charge due to the quaternate chitosan’s cationic group, that favor the interaction with the negatively charged bacterial cell walls. This system reduced the wound area in rats up to 41.3% after 7 days, decreased inflammation, and improved re-epithelialization [27].

In a similar study, Xie et al. [22] prepared an Ag/chitosan hydrogel using an alkali–urea solution, LiOH (4.5% wt.)/KOH (7% wt.)/CH_4_N_2_O (8% wt.) by the freeze/thaw process, AgNO_3_, and Na_3_C_6_H_5_O_7_. The silver concentration in the hydrogel increased, leading to spherical and ellipsoidal Ag NPs with a size distribution of 4.45 nm ± 0.37 nm to 9.22 ± 0.54 nm. The hydrogel composite had large tensile mechanical properties (15.95 ± 1.95 MPa). The antimicrobial activity was 99.86 ± 0.12% against *E. coli* and 99.94 ± 0.10% against *S. aureus* tested on rats for 14 days. The wound contraction was 70.5% on the 4th day and 99.75% on the 14th day. Thus, Ag NPs coated with chitosan accelerated the healing process. The authors determined that Ag NPs destroyed the bacterial cell wall due to interactions between the NPs and the lipid layer of the bacterial cell membrane. The Ag NPs would merge with bacterial DNA damaging bacterial replication and impairing bacterial respiratory function.

Furthermore, carboxymethyl chitosan is a derivative of chitosan, non-toxic, and also capable of forming gels [62]. Carboxymethyl chitosan-based hydrogels have shown enhanced physicochemical, and biological properties, including antimicrobial, antioxidant, and antifungal activities. This hydrogel has been used in applications such as wound healing, drug-carrying, smart tissue, and biomedical nanodevices [106]. Additionally, it has been well explored in the cosmetic and food industry [62].

Ag/chitosan-carboxymethyl β-cyclodextrin hydrogel (CM-βCD) is an alternative approach to inhibit the growth of bacteria. It has been shown to display antibacterial activity against *E. coli* and *S. aureus* [25]. The interactions between Ag^+^ ions and bacteria were improved through ions exchange between Ag^+^ and H^+^ from the carboxylic and amino groups within the Ag NPs-CM-βCD hydrogel. The inhibition zone increased when the concentration of CM-βCD was increased in the hydrogel [25].

Pandian et al. [37] fabricated a Ag/N, O-carboxymehtyl chitosan (N, O-CMC) hydrogel with self-healing properties. The ethylenediaminetetraacetic acid (EDTA, C_10_H_16_N_2_O_8_) and ferric ions (Fe^3+^, FeCl_3_, 2%) were used in the synthesis process to produce a self-healing hydrogel. The size distribution of Ag NPs was 25 ± 14 nm according to TEM images. The hydrogel displayed an antibacterial activity against ATCC and clinical strains of *E. coli* ATCC 25922, *S. aureus* ATCC 35556, MRSA ATCC 43300, *P. aeruginosa* ATCC 47085, and *K. pneumonia* ATCC 700603. The minimum inhibitory concentration (MIC) for *P. aeruginosa* was 48.5 mg/mL, 32.5 mg/mL for MRSA, and 32 mg/mL for *S. aureus*. The Ag NPs/N, O-CMC hydrogel was more efficient against *E. coli* and *K. pneumonia* with MIC values of 17.5 and 23.0 mg/mL, respectively. At the same time, the minimum bactericidal concentration (MBC) values were 55 and 71 mg/mL, respectively. The authors described that the interaction between Fe^3+^ (metal) and -COOH (ligand) was responsible for the self-healing property of the Ag NPs/N, O-CMC hydrogel [37].

The carboxymethyl chitosan (CMCS) has been mixed with oxidized konjac glucomannan (OKGM). The OKGM is a natural polysaccharide, soluble in water, that was shown to improve the microstructure and mechanical properties of chitosan [107,108], gelatin [109], and oxidized hyaluronic acid [110], acting as a macromolecular cross-linker [108]. The OKGM-based hydrogel exhibited self-healing characteristics in a recent study, where Ag NPs/OKGM/CMCS hydrogel demonstrated antibacterial properties against *E. coli* and *S. aureus* [23]. This hydrogel was tested on rats’ skin. The hydrogel pore size distribution was in the range of 59.4 to 230 µm, increasing as the concentration of OKGM increased, but the swelling capacity decreased. Higher concentrations of polymers accelerated the gelation time from 600 to 57 s [23]. Similar to a previous study, Ag/konjac glucomannan hydrogel was tested against *S. aureus* and *E. coli* showing good antibacterial efficiency on rabbit skin infections [111].

Hydrogels based on carboxymethyl cellulose (CMC), polyvinyl alcohol (PVA), and C_8_H_14_O_4_ (EDGE) has been prepared using microwave radiation as a carrier of Ag NPs (8–14 nm). The Ag release rate from this hydrogel was 85% over five days [13]. Ag^+^ ions are bound to the hydrogel composite via electrostatic interactions. This Ag/hydrogel acted as a bactericide against pathogenic microorganisms of the urinary tract, such as *E. coli*, *K. pneumoniae*, *P. aeruginosa*, *P. vulgaris*, *S. aureus*, and *P. mirabilis* [13]. The Ag/hydrogels with 5 mg/mL of Ag presented a growth inhibition diameter of 16.6 mm against *E. coli*, 15.8 mm against *K. pneumoniae*, 15.6 mm against *P. aeruginosa*, and 15.2 mm against *P. vulgaris*.

Hydrogels based on carbopol-934 and *Aloe vera* supported Ag spherical NPs encapsulating quercetin (QCT) [63]. This system was designed to take advantage of (i) the properties of QCT as an anti-inflammatory and antioxidant; (ii) of carbopol-934, as a biodegradable and bioadhesive polymer with good tensile strength; (iii) *Aloe vera* that stimulates collagen production; and finally, (iv) of Ag NPs that have broad antimicrobial activity. The QCT-Ag/carbopol-*Aloe vera* hydrogel presented antibacterial activity against *S. aureus* MTCC 3160 and *E. coli* BL-21 with inhibition zone values of about 19.0 and 17.0 mm, respectively. Ag NPs improved the release rate of quercetin from the hydrogel for the treatment of wounds in diabetic patients.

Some studies have explored the incorporation of graphene into hydrogel structures due to its high thermal and electrical conductivity, and sizeable mechanical strength [66,70,71,72]. The graphene embedded in hydrogel reduced hydrogel breaking and reinforced its mechanical properties. The Ag/graphene composite hydrogel was prepared using acrylic acid and N,N′-methylene bisacrylamide (C_7_H_10_N_2_O_2_), with a mass ratio of 5:1 silver to graphene [41]. The Ag NPs of an average size of 39 nm were deposited onto the surface of graphene nanosheets. The Ag NPs/graphene hydrogel was evaluated against *E. coli* and *S. aureus* using the shaking flask method. The antimicrobial activity was enhanced as the Ag NP concentration increased. Larger nanoparticle sizes displayed better antimicrobial activity than smaller ones. The graphene promoted the incorporation of a higher number of NPs and avoided their aggregation onto its surface.

Another approach that has been explored is to combine chitosan and graphene to produce an antibacterial hydrogel with enhanced durability [71]. For instance, Nešović et al. prepared Ag/poly(vinyl alcohol)/chitosan/graphene hydrogels [70,71,72] by electrochemical synthesis of nanoparticles in a hydrogel network. The hydrogel displayed better mechanical characteristics, such as tensile strength and elastic modulus. The Ag NPs size distribution was from 6.38 to 10.00 nm depending on the chitosan content. The antimicrobial activity was evaluated against *E. coli* ATCC 25922 and *S. aureus* TL. The number of bacteria colonies decreased quickly in 15 min, when the AgNO_3_ concentration was 0.25 mM and 0.5% wt. of chitosan, during the composite hydrogel preparation (0.25Ag/PVA/0.5CHI/Gr). Increasing the chitosan content resulted in a slower Ag release rate from the hydrogel. Nešović et al. [71] found that Ag NPs prevented adenosine 5′-triphosphate (ATP) formation within the microorganism.

Figure 1 summarizes the hydrogels embedded with Ag and Au NPs against multidrug-resistant bacteria.

A Ag-polyethyleneimine (PEI)-graphene oxide (GO) hydrogel was produced using Pluronic F127 gel [66]. In this case, Pluronic F-127 was used to create a sustained antimicrobial effect, presenting reverse thermal gelation properties. PEI decreased the aggregation of nanostructures within the hydrogel. The antimicrobial activity against *E. coli* was 99.86%, and 99.94% against *C. albicans*, using 10 µg/mL of the hydrogel. The Ag release rate from the hydrogel was 72% in 7 days. The authors proposed that the graphene oxide nanosheets damaged the bacterial cell wall due to the sharp edges leading to a faster disruption of the plasmatic membrane by the Ag NPs.

Furthermore, Ag NPs have been incorporated into nanotubes/polymer hydrogels to explore NP delivery. For instance, aluminosilicate nanotubes (NTs) are platforms with a great capacity to store and carry molecules and drugs. They also help to reduce the hydrogel degradation rate and can be loaded, as additives, into the hydrogels, such as gelatin methacrylate (GelMA), a biocompatible hydrogel with many biological characteristics [112]. For example, Ag NPs were loaded into aluminosilicate nanotubes (NTs) and then within a methacrylate gelatin [65] matrix, to produce an antibacterial hydrogel capable of improving bone regeneration. The hydrogel was prepared using photopolymerization by UV irradiation of 365 nm and 400 W. According to the inhibition zone results, the Ag/NTs/GelMA hydrogel showed higher antibacterial activity against *E. coli* ATCC 8739 than *S. aureus* ATCC 29213.

In addition, the morphology of Ag NPs is another relevant aspect that influences a hydrogel’s antibacterial efficiency. Different NP shapes may present a distinct surface area to interact with bacterial membranes, leading to diverse antibacterial activity [40,113,114]. In this context, Ag NPs with different morphologies (spherical, triangular, and rod) were incorporated into polyacrylamide (PPA) and N-mehtylene bisacrylamide (MBA) hydrogels, named PAA-MBA [40]. The mechanical strength of the Ag NPs-PAA-MBA hydrogel (4 to 5 KPa) did not depend on Ag NP shape. Rod-shaped nanoparticles were poorly absorbed within the hydrogel network due to the formation of aggregates on the hydrogel surface. However, these NPs showed antibacterial activity. The hydrogel doped with spherical NPs of 12.7 ± 5.9 nm and triangular NPs of 37.1 ± 15.0 nm demonstrated high antimicrobial activity against *E. coli.*

Table 2 summarizes the hydrogels embedded with Ag NPs for antibacterial application.

### 2.2. Antibiofilm Activity of Hydrogels Loaded with Ag NPs

Taking the usefulness of non-invasive therapy into consideration, and the elimination of drug-resistant biofilms in oral infections and wound healing, hydrogels loaded with Ag NPs is an alternative method of infection management [115,116,117]. In this scenario, Haidari et al. [43] investigated the effectiveness of applied Ag NP hydrogels in mature *S. aureus* biofilms, both in vitro and in vivo. In vitro tests were performed by flow cytometry, where bacterial cells with compromised membranes were stained red by propidium iodide, whereas cells with intact membranes were stained green by SYTO9. The test showed that after treating the *S. aureus* biofilm with the Ag NP hydrogel, most of the cells were stained in a high red fluorescence intensity, associated with a substantially lower biofilm biomass, indicating severe disruption of the mature biofilm. For in vivo tests, an established *S. aureus* mouse model of a mature biofilm wound infection was utilized. The antibiofilm treatment started after biofilms had been fully established. IVIS bioluminescent imaging was used to track 10 days of Ag NP hydrogel treatment in real-time. The Ag NP hydrogel treatment gradually decreased the *S. aureus* biofilm starting on day 4.

From 5 to 10 days after the infection, there was a statistically significant decrease in the concentration of bacterial cells, showing the high efficiency of the Ag NPs in eradicating established mature biofilms in wounds. This study demonstrated the use of an Ag NP hydrogel as a valid therapeutic approach for the effective and safe elimination of mature *S. aureus* biofilms in wounds.

Consistent with these results, Imran et al. [118], also reported the antibiofilm activity of a Ag NP-loaded hydrogel against *B. subtilis* and *E. coli*. It was revealed that the hydrogel showed a dose-dependent biofilm inhibition activity, with a minimum biofilm inhibition of approximately 27% when the Ag NPs were used at a concentration of 10 ppm and a maximum inhibition of 97% when the Ag NPs were used at a concentrations of 100 ppm. Additionally, the half maximal inhibitory concentration (IC_50_) values obtained were 29.88 and 27.36 for *E. coli* and *B. subtilis*, respectively. Pandian et al. [37], in turn, evaluated the antibiofilm activity of in situ Ag NPs incorporated in an N, O-carboxymethyl chitosan self-healing hydrogel. After 48 h, a decrease of 68.86 ± 0.05%, 75.07 ± 0.02%, and 83.22 ± 0.01% was observed in *E. coli-*, *S. aureus-*, and *P. aeruginosa*-treated biofilms, respectively.

Alfuraydi et al. [119] described the preparation of novel cross-linked chitosan and PVA hydrogels impregnated with Ag NPs, as well as its activity against different strains of fungi, Gram-positive and Gram-negative bacteria. In their results, The minimal biofilm inhibition concentration (MBIC) for the chitosan hydrogels alone ranged from 15.63 to 125 µg/mL, differing from the MBIC values of the hydrogel containing Ag NPs at 1 and 3%, which ranged from 1.95 to 7.81 µg/mL. These data demonstrated how the dispersion of Ag NPs inside the matrix of the chitosan hydrogel significantly improved its ability to prevent the formation of biofilms.

Similarly, the antibiofilm action of the chitosan hydrogel containing Ag NPs was previously explored by Pérez-Díaz et al. [120]. In their work, the hydrogels demonstrated a great impact on the multi-species biofilm of oxacillin-resistant *S. aureus* (ORSA), achieving a 6 Log_10_ reduction at a Ag NP concentration of 100 ppm. The antibiofilm activity against *P. aeruginosa* was lower, with a Log_10_ decrease of 3.3 at a concentration of 1000 ppm. As stated in the study conducted by Arinah et al. [121], the different results on the tested drugs’ antibiofilm activity could be attributed to structural variations in the bacterial membrane walls, which differ in Gram-negative or Gram-positive bacteria. In their work, the authors incorporated *Pleurotus ostreatus*-biosynthesized Ag NPs into a genipin-crosslinked gelatin hydrogel to investigate the antibiofilm properties against the biofilms of *S. aureus*, *P. aeruginosa*, *Bacillus* sp., and *E. coli*. Stronger biofilm inhibition of about 58 ± 4% was observed in Gram-negative strains. For Gram-positive bacteria, the percentage of inhibition was 55 ± 5% for *S. aureus* and 38 ± 1% for *Bacillus* spp.

Furthermore, many recent studies have reported antibacterial and antibiofilm activity improvement of drugs when they are associated with metallic nanoparticles, such as Ag NPs [122,123]. Thus, in the research conducted by Lopez-Carrizales et al. [124], chitosan hydrogel loaded with Ag NPs and the antibiotic ampicillin (AMP) were tested against resistant bacterial pathogens, evaluating its capacity to prevent the early formation of biofilms by the colony biofilm model. The biofilm produces thick, layered structures, and the counting of colony-forming units (CFU) was Log_10_-transformed. The antibiofilm action of the hydrogel changed depending on the Ag NP and ampicillin concentrations and the tested strain. The biofilms of *A. baumannii*, *E. faecium*, and *S. epidermidis*, were significantly inhibited by the hydrogel with the lowest concentration of Ag NPs and ampicillin (25 ppm Ag NPs/50 ppm AMP), exhibiting Log_10_ reductions of 10 ± 0.01, 8.9 ± 0.02, and 7.8 ± 0.13, respectively. However, the *E. cloacae* biofilm was only inhibited by a higher antimicrobial dose (250 ppm Ag NPs/500 ppm AMP), resulting in a Log_10_ reduction of 9.9 ± 0.11.

Recently, Wunno et al. [125] investigated a potentially new sustainable delivery system of Ag NPs for, among other activities, antibiofilm action. In their work, an ex situ thermosensitive hydrogel based on poloxamers loaded with biosynthesized Ag NPs from *Eucalyptus camaldulensis* was created and tested against Gram-positive (*S. aureus* and *S. epidermidis*) and Gram-negative bacterial (*A. baumannii* and *P. aeruginosa*) biofilms. At a ½ minimum inhibitory concentration (MIC), the proportion of biofilm inhibition reached 83%. When the mature biofilms were exposed to the Ag NP hydrogel and analyzed by confocal laser scanning, loosening of the biofilm architecture and cell death were revealed after 4 h of co-incubation with the hydrogel formulation at a 2 MIC (μg/mL) concentration. Based on the presented results, it is clear that the tested hydrogel formulation successfully interrupted biofilm formation and eradicated cell viability within the mature biofilms.

## 3. Gold Nanoparticles (Au NPs)

The wide range of applications of Au NPs is related to their physicochemical properties such as the tunable core size, photothermal [126] and photodynamic properties, high chemical stability, biocompatibility [127], high X-ray absorption coefficient, efficiency in generating ROS, and localized surface plasmon resonance (LSPR) properties [92,128,129]. Furthermore, Au NPs also exhibit antimicrobial properties absent in bulk or ionic gold. For example, Au NPs destroy bacterial membranes and slow down their metabolism [58,93,94,95]. Due to the NP’s small size, gold colloid may be susceptible to NP aggregation. Thus, Au NPs are usually stabilized with additives such as polyelectrolytes or polymers [96,97]. These stabilizers act as capping or protecting agents, and they prevent aggregation due to steric hindrance [97,98,99,100,101,102,103,104,105,106,107].

In the design of advanced hydrogels, the Au NPs are embedded into a hydrogel, or in situ synthesized inside the porous gel structure. Some of the polymers used to incorporate Au NPs, include chitosan [60], alginate [59], gelatin [61], hydroxypropyl methylcellulose [24], silk [26], acrylamide, diethylene glycol, and indole-3-acetic acid, poloxamer 407, Pluronic F-127, carbopol [64], carboxy methyl tamarind, methacrylated gelatin, and metal-organic frameworks (MOFs) [69]. Au NPs encapsulated in hydrogels have shown antimicrobial or bactericidal activity against Gram-positive bacteria, such as *Bacillus cereus*, *S. aureus* [126], and *S. epidermidis* [124]. Additionally, against Gram-negative bacteria such as *P. aeruginosa* [61,126], *E. coli* [59,126], *K. pneumoniae*, and *E. cloacae* [29], and fungus such as *C. albicans* [24].

Some important characteristics of Au NPs, such as size and shape have been tailored and explored to improve the antibacterial activity of the hydrogel. This was evidenced in studies that developed nanospherical of 29.2 nm [130], nanorods of 82.5 nm [24], 54 nm [131], 49.2 nm [130,132], and nanostars with a core diameter of 25 nm and an average size of 50 nm, 70 nm, and 120 nm [126]. The latest advances in gold nanoparticles embedded in hydrogels for the treatment of multidrug-resistant bacterial infections are discussed below. The efficiency of Au NPs against pathogenic bacteria is presented from three relevant aspects: antibacterial activity, biofilm activity, and antibacterial and antibiofilm mechanism of action.

### 3.1. Antibacterial of Au NPs Loaded into Hydrogels

Au NPs have shown very promising results against multi-resistant bacteria to antibiotics. The incorporation of Au NPs into biocompatible supports, such as liposomes, is one approach used in biomedicine. This structure can interact easily with bacterial membrane and possesses a high-delivery capacity for NPs, antibiotics, enzymes, etc. To treat bacterial infections, Zhang et al. [133] fabricated a hydrogel containing pH-responsive gold nanoparticle-stabilized liposomes as a topical antimicrobial carrier. The authors used carboxyl-modified AuNPs as stabilizers for cationic liposomes and chemically cross-linked polyacrylamide as a hydrogel. The hydrogel viscoelasticity was tailored by the cross-linker concentration, and this resulted in tunable release kinetics of the Au NP liposomes. *S. aureus* was used as a model pathogen, and the hydrogel formulation effectively released nanoparticles into the bacterial culture. No skin reaction was observed when the hydrogel formulation was topically applied to mouse skin over a 7-day treatment period [133].

One of the methodologies used to obtain hydrogels uses natural polymers, such as alginate. Alginate is a hydrophilic linear polysaccharide extracted from the cell wall of some specific species of algae or bacteria. Alginate can form a gel or act as a crosslinker with other polymers due to the exchange of Na^+^ ions from the guluronic acids (C_6_H_10_O_7_) with other cations (Ca^2+^, Ba^2+^, and Mg^2+^) [134]. Alginate-based hydrogels are biocompatible, biodegradable, non-toxic, and exhibit a higher capacity of fluid load, acting as a carrier for NP delivery [135]. Gold nanostars (Au NSts) were loaded into sodium alginate-based hydrogel by Kaul et al. [126]. The sharp spike (size of 120 nm) from the NSts could puncture the bacterial wall and membrane. The antimicrobial activity was 35.4% against *S. aureus* MTCC 1430, as higher as 80% against *P. aeruginosa* MTCC 1934 and *E. coli.* MTCC 443, using 0.3 to 0.6 µg/mL of nanoparticles on the wounds of Sprague Dawley rats. The spike length, as well as the topology, of the Au NSts damaged the surface and the bacterial membrane due to the rupture process. *S. aureus* were more resistance due to the thick peptidoglycan layer outside the bacterial cell wall. In similar study, Zhang et al. [59] prepared an acrylamide (AM) and alginate (SA) hydrogel incorporating Au NPs (8 nm). This Au/AM-SA hydrogel inhibited the growth of *E. coli*. The study suggested that Au NPs interact with the capsule of *E. coli*, cross the cell wall, and attack the proteins of the membrane and cell wall. This process leads to the disruption of the outer membrane, followed by death of the *E. coli.*

Other biocompatible and biodegradable polymers have been used as a base to produce advanced hydrogels with metallic NPs, such as chitosan (CS) and gelatin. Lu et al. [60] synthesized Au NPs in chitosan solution. Then, their surface was functionalized with a shell of 2-mercapto-1-methylimidazole (MMT), resulting in an Au-CS–MMT nanocomposite with size 8 to 10 nm. This system was loaded into gelatin using tannin acid as a crosslinker, the final product was Au-CS–MMT/gelatin. The antimicrobial activity was explored against *S. aureus* ATCC 25923, *E. coli* ATCC 25922, and MRSA using New Zealand rabbits as a model. The minimum inhibitory concentration (MIC) was <20 µM for the three bacteria strains. The antibacterial activity of Au-CS–MMT/gelatin was better than the standard ampicillin treatment used as a control. The results from the surface charge by zeta potential and the content of Au in *E. coli* and *S. aureus*, confirmed the strong electrostatic interactions with the Au-CS–MMT particles. Additionally, scanning electron microscopy (SEM) and transmission electron microscopy (TEM) images showed that Au-CS–MMT damaged the morphology and disrupted the bacterial cell membrane in less than 1 h of contact. Ryan et al. [136] synthesized a chitosan and siloxane hydrogel to incorporate Au NPs. Tetraethyl orthosilicate (TEOS) was used to form an interpenetrating polymer network and improved the hydrogel structural properties, such as flexibility and strength (67–74 mPa). The size distribution of the Au NPs was 19 nm ± 18%. Antimicrobial tests displayed that cross-linking with SiC_8_H_20_O_4_ (TEOS) reduced the attachment of *E. coli* to the well plate surface by 80%.

Gelatin is a natural, amphoteric, non-inflammatory polymer and it is obtained from the hydrolysis of collagen [60]. It has many functional groups allowing its polymerization with several crosslinking agents [137]. A gelatin-based hydrogel was fabricated by Jiang et al. [61], in which Au NPs (5 nm) were capped by 6-aminopenicillanic acid (APA), and embedded into electrospun fibers of poly(ε-caprolactone)/gelatin. The MIC of Au–APA/gelatin was 2.5 µg/mL against *E. coli* and *K. pneumoniae*, >5 µg/mL against *P. aeruginosa*, 5 µg/mL against MDR *E. coli* and MDR *K. pneumoniae*. It was observed that *E. coli* cell walls were leaky and broken when the concentration of Au–APA/gelatin increased.

Hydroxypropyl methylcellulose (HPMC) is a non-toxic and non-ionic biopolymer which is used as a stabilizer, thickener, and emulsifier in several applications in the food and pharmaceutical industry. HPMC has many polar and non-polar groups which easily interact with nanoparticles by coordination bonds [138]. Recently, Wafaa Soliman et al. [24] obtained embedded Au rod-shaped NPs into a HPMC hydrogel for topical application. The size distribution and surface charge (ζ) of the Au NPs was 82.5 nm and 34.8 mV, respectively. Male Wistar rats were used as a model for in vivo studies. The MICs against *S. aureus* ATCC 10400, *E. coli* ATCC 25922, and *C. albicans* ATCC 90028 were 0.125–0.25 ng/mL. The minimum bactericidal concentrations (MBCs) were 0.1–0.5 ng/mL. This hydrogel was more efficient against *S. aureus* and *E. coli*. The authors suggested that the interactions between Au NPs and bacteria happened by the large cationic surface charge of the Au nanorods leading to membrane disruption, damage to the bacterial cell structures, and consequently death of the pathogenic microorganism.

A treatment for focal infections, based on laser-mediated heating of Au NPs (13 nm) suspended in an injectable and degradable silk hydrogel, has also been suggested [26]. Silk is a natural, biocompatible, cheaper polymer, and the silk hydrogel characteristics can be tailored and controlled by the gelation time [28,139]. The bactericidal procedure consists of injecting the silk hydrogel/Au NPs composite into the subcutaneous infection, and to deliver a laser beam with 150 mW of incident green light (532 nm wavelength) for 10 min [26]. The wavelength light is absorbed by the nanoparticles and converted into heat. This localized heat has a bactericidal effect at the infection site without causing systemic side effects. The in vivo results showed *S. aureus* reduced after one round of laser-exposure, killing 80% of bacteria, demonstrating the potential applicability of this proposal.

In addition to hydrogels obtained from natural polymers, other types of synthetic polymers or monomers have been used for hydrogel preparation for NPs delivery. A pH-sensitive hydrogel with antimicrobial activity and wound-healing properties was produced by Chitra et al. [140]. Au NPs of 17 nm were loaded into a porous hydrogel, obtained by condensation–polymerization reactions with citric acid (CA), diethylene glycol (DEG), and indole-3-acetic acid (IAA, C_10_H_9_NO_2_). The swelling profile of the hydrogel decreased when the content of Au NPs increased, in basic medium. The antibacterial performance against *S. aureus* showed an inhibition zone of 8.33 to 11.67 mm, using 1000–2000 µg/mL of Au/hydrogel by the diffusion method. In a similar study, Au NPs (8–30 nm) and Ag NPs (4–12 nm) were incorporated into a hydrogel composite based on the condensation–polymerization between citric acid (CA), diethylene glycol (DEG), and indole-3-acetic acid (IAA) [128]. This hydrogel nanocomposite was tested against *S. aureus*, *E. coli*, and *Bacillus cereus* at 2000 µg/well. The inhibition zones of the Au/hydrogel were 14, 16, and 15 mm against *E. coli*, *S. aureus*, and *B. cereus*, respectively. Chitra et al. [128] described that the results were due to the outside structure of Gram-positive bacteria, which allows the entry and absorption of foreign molecules into the bacterial cell membrane.

Poloxamer 407 is a triblock copolymer (poly(ethylene glycol)-block-poly(propylene glycol)-block-poly(ethylene glycol)). It is water-soluble, and has been explored in the pharmaceutical industry as an antibiotic delivery platform [141]. A polymeric hydrogel with Au NPs was reported by Mahmoud et al. [130] to treat wounds. A poloxamer 407 hydrogel was used to support sphere- and rod-shaped Au NPs with different coating agents, such as CTAB (C_16_H_33_N(CH_3_)_3_Br), polyacrylic acid (PAA), poly(allylamine hydrochloride) (PAH), and poly(ethylene glycol) (PEG), to endow the surface with negatively, positively and neutrally charged polymers. Rod-shaped Au NPs capped with PEG and positively charged NPs capped with PAH proved to be the most efficient systems for wound healing after 14 days of treatment. Likewise, the two hydrogel systems presented a high reduction in viable bacterial against *S. aureus* and *P. aeruginosa*, the most common skin bacteria.

Pluronic F-127 is a synthetic thermoresponsive polymer which displays sol–gel transition near 37 °C, excellent biocompatibility, good mechanical strength, and the ability to retain water [142]. An injectable hydrogel for muscle regeneration was produced by Ge et al. [68], in which gold–polythyleneimine NPs (10 nm) were embedded into a Pluronic F-127 hydrogel scaffold, named FPAu. This FPAu biomaterial was obtained by the double crosslinking of Pluronic F127, 4-hydroxy benzaldehyde, K_2_CO_3_, and modified polydopamine NPs. The number of colony-forming unit decreased rapidly after 2 h of contact between bacteria and the hydrogel in in vitro tests. The antibacterial activity against *S. aureus* and *E. coli* was 87.5% and 83%, respectively. This study suggested that the antibacterial property of the FPAu hydrogel is due to branched polyethyleneimine linked to the Au NP surface.

Another approach is to coat the hydrogel with a pretreated macrophage membrane of bacteria. This creates a bacterial receptor able to identify specific sites when it interacts with the target pathogenic bacteria. Li et al. [131] fabricated a photothermal hydrogel of N-acryloyl glycinamide (PNAGA) in which Au nanorods were previously coated in polydopamine (PDA). The hydrogel was also coated with membrane macrophages against *E. coli* and *S. aureus*. The PNAGA enhanced the mechanical properties of the hydrogel, showing a tensile strength of 1.64 MPa and a compressive strength of 12.490 MPa. Au nanorods gave a photothermal ability to the hydrogel under NIR irradiation. When exposed to NIR irradiation for 5 min, the antibacterial activity of this hydrogel, without the macrophage membrane, was 74.2 and 72.5% against *E. coli* and *S. aureus*, respectively. However, the hydrogel coated with the activated membrane of macrophage against *E. coli* and *S. aureus*, led to an antibacterial efficiency of 98.4 and 97.6%, respectively.

Some authors have explored alternative methodologies, for example, by incorporating dual metallic nanoparticles, such as Ag and Au NPs, as core–shell nanoparticles into the porous structure of a carbopol-based hydrogel [64]. The particle size was 5 ± 3 nm and the Ag–Au NPs presented several morphologies into the hydrogel. The inhibition zones against *B. cereus* and *E. coli* were 18.5 and 18.1 mm, respectively. This study showed that Ag–Au NPs inhibited the growth of bacteria by forming pits between the NPs and the cell wall. This interaction resulted in bacterial death due to leakage of molecules and proteins from the wall.

The bimetallic NPs, Au–Ag NPs were incorporated by Kumar et al. [29] in a carboxy methyl tamarind (CMT) hydrogel against MDR *E. coli*, *E. cloacae*, and *S. aureus* MRSA for in vitro tests using mammalian cells. The bimetallic NPs had a hydrodynamic size of 147 nm and a negative surface charge of −31.5 mV. The growth profiles of the cells were studied at different concentrations of Au–Ag NPs. *E. coli* showed an extended lag phase when exposed to Au–Ag NPs at a concentration of 1 to 3 µg/mL, while at a concentration of 3 to 12 µg/mL for MRSA in presence of Au–Ag NPs. The lag phase is the earliest period of the bacterial growth cycle, which the bacteria adjust to their environment, and cells increase in size [143]. The reported MIC values were 3 and 6 µg/mL for *E. coli* and MRSA, respectively. This hydrogel was also tested against clinical isolates of *E. cloacae* EC18, which was efficient at 6 µg/mL. The antimicrobial activity results showed that the Au–Ag/hydrogel was efficient when 20- to 25-fold less concentrated than other drugs, such as gentamicin.

In a similar study, a gelatin sponge hydrogel functionalized with silver/gold clusters (Au/Ag–gelatin and Au–gelatin) was used for antibacterial applications [144]. This system was obtained by a simple one-pot method. Glutathione (GSH) acted as a reducing agent and as a thiol-ligand. The system’s biocompatibility, as well as good water absorbency and water retention properties, allows efficient bactericidal effects and presents this hydrogel as a promising material for wound dressing applications. The antibacterial activities of gelatin, Au–gelatin and Au/Ag–gelatin were probed by inhibition zone assays, using *P. aeruginosa* as a model, since it is implicated in wound infection. Gelatin did not present any inhibition zone under any of the conditions, while for Au–gelatin and Au/Ag–gelatin, the inhibition appeared under white light irradiation. According to the authors, the bactericidal activity is due to ROS generated by the excited NCs. The inhibition zone of Au/Ag–gelatin was 31.9 mm higher than the Au–gelatin with 25.1 mm.

Ribeiro et al. synthesized in situ Au and Ag NPs embedded in a silk fibroin-based hydrogel [28]. The size distribution of the Au NPs was 9 to 55 nm and 12 to 69 nm for the Ag NPs. The NP concentration influenced the antimicrobial activity. For example, the hydrogel loaded with a Au NPs concentration > 0.5% was efficient against the *S. aureus* ATCC 33591, MRSA, and *P. aeruginosa* ATCC 27853. When the concentration was >0.1%, it inhibited the cell proliferation of *S. aureus* ATCC 25923 and MSSA, and *E. coli* ATCC 25922. However, the hydrogel showed antimicrobial performance against all previous pathogenic cells, regardless of Ag NPs concentration, even with *S. epidermidis* RP62A ATCC 35984.

Furthermore, Au NPs have been loaded into porous systems such as metal–organic frameworks (MOFs). The zeolitic imidazolate framework-8 (ZIF-8) MOF are crystalline, biocompatible, and biodegradable. ZIF-8 can generate ROS under visible light by means of photocatalysis [145]. The Au NPs/ZIF-8 strategy can improve the antimicrobial activity of Au NPs. For example, Deng et al. [69] embedded Au NPs into a pristine ZIF-8 network. The nanocomposite was embedded in oxidized sodium alginate (OSA), and carbohydrazide-modified methacrylated gelatin (GelMA-CDH) obtaining an injectable hydrogel, named Au–ZIF–GCOA. The stability of the hydrogel was improved by adding another crosslinking step during the polymerization, leading to a double-network hydrogel. This material displayed high bactericidal activity against *E. coli* and *S. aureus* using mice as an in vivo model. The number of bacteria colonies decreased in more than 99.0% for both strains using 0.2 mg/mL of Au–ZIF–GCOA. The antibacterial activity was due to the photoproduction of hydroxyl radicals by Au–ZIF-8 nanostructures, under visible-light irradiation. Mainly, Au NPs were responsible for the conversion of oxygen to singlet oxygen (^1^O_2_). The interaction of the hydrogel with light prevented the bacteria from acquiring resistance mechanisms to metal nanoparticles.

Other strategies developed against bacteria resistance combine the properties of metallic nanoparticles conjugated with drugs in the same hydrogel structure. Au NPs were incorporated into a cellulose-grafted polyacrylamide hydrogel [146], leading to a PAMC/Au nanocomposite. Afterwards, ciprofloxacin was embedded into this nanocomposite/hydrogel. The antibacterial performance was evaluated against *E. coli*, *S. flexneri*, *B. cereus*, and *Listeria Inuaba.* An increased concentration of Au NPs into the hydrogel was observed to enhance the antibacterial activity from 67 to 95% against the *E. coli*, and from 48 to 79% against the *S. Flexneri*. However, this hydrogel nanocomposite was not as efficient against *B. cereus* and *L. Inuaba* with an antimicrobial activity only between 35–53%. The study suggested that antimicrobial efficacy was influenced by the hydrogen bonding interaction between the Au NPs and the amide acrylamide groups.

Au NPs have been incorporated into the hydrogel network combined with Ag NPs leading into antibacterial hydrogels with dual metallic NPs (Au–Ag). Table 3 summarizes the hydrogels containing Au NPs for antibacterial application. The antibiofilm activity of Au NPs incorporated into hydrogels is shown below.

### 3.2. Antibiofilm Activity of Au NPs Loaded into Hydrogels

Au NPs have previously shown promising results against several microorganisms and biofilms in terms of growth inhibition and cell damage [147,148]. With this background information, Bermúdez-Jiménez et al. [149] embedded gold nanorods (Au NRs) into a non-toxic chitosan hydrogel, exploring its antibiofilm activity against Gram-positive and Gram-negative pathogenic bacteria multi-species biofilms, by photothermal therapy. The authors reported 5 to 8 Log_10_ reductions in bacterial load when the *Streptococcus oralis* and *E. faecalis* biofilm was exposed to the Au NR hydrogel subjected to a 10 °C temperature rise. However, when the hydrogel was subjected to a 5 °C temperature rise, no discernible drop in the bacterial load was seen. These findings suggest that photothermal therapy was essential in antibiofilm activity of this Au NR-loaded hydrogel. Correspondingly, Al-Bakri et al. [132] also investigated the potential of Au NRs incorporated into a hydrogel by photothermal therapy against *P. aeruginosa* biofilms. In this study, the photothermal-based bactericidal activity of the Au NR hydrogel against biofilms showed the same percentage and Log reduction in viable bacterial count under two different modes of laser excitation. The results showed approximately 4 Log cycle, and 1 Log cycle reduction in the viable cells for both, continuous and pulsed laser excitation at 3 W cm^−2^ and 1 W cm^−2^ laser doses, respectively.

Wickramasinghe et al. [150] also investigated the photoactivated Au NRs incorporated into hydrogel composites, to explore their antibiofilm activity against *S. aureus* bacterial biofilms on metal implant materials. A set of 1 W/cm^2^ power intensities, with a 1 cm^2^ laser spot size, and 15 s of laser pulses was devised to assess the hydrogel’s ability to completely eradicate preformed bacterial biofilms on the surface of metal alloy disks. According to crystal violet assays, the hydrogel completely eradicated the biofilms. Additionally, a colony-forming assay was carried out. The results demonstrated that no surviving bacteria established new colonies because of the gel treatment. Finally, SEM analysis showed that this hydrogel did not leave any bacterial cells on the surface of the metal alloy disks, even in the microscale grooves. SEM analysis conducted by Soliman et al. [24] using *S. aureus* and *E.coli* biofilms has also shown the intense reduction in cell number and morphological changes after treatment with Au NPs incorporated into hydrogels.

Recent work also sought to depict this technique as a useful strategy against biofilm formation from different bacterial species. In 2022, Galdámez-Falla et al. [151] developed an *E. faecalis* biofilm on human roots using the static and dynamic method (modified drip flow reactor (MDFR), aiming to use photothermal therapy with a hydrogel solution with Au NRs as an antibiofilm agent. The authors found differences in colony-forming unit (CFU) when comparing the Au NRs-treated biofilm (188.6 ± 26.7 CFU) with the control group (337.3 ± 2.82 CFU). The Au NRs successfully eliminated *E. faecalis* biofilms. The laser application time, however, was 20 min, which is longer than would be feasible for an in vivo scenario. The researchers suggested that, in the future, this strategy should be tested with a shorter laser application time with similar positive outcomes.

Another antibiofilm strategy that has been explored used Au NPs hydrogels in association with Ag NPs. Due to the effectiveness in combating bacteria, this strategy has received great attention among many researchers. The synergistic effect is well-described by Kumar et al. [29]. In their study, bimetallic NPs (Au–Ag NPs), capped with complex carbohydrates, and incorporated into a carboxy methyl tamarind (CMT) polysaccharide hydrogel, showed a dose-dependent effect on biofilm formation for both Gram-positive and Gram-negative bacteria. Low concentrations, such as 1 μg/mL for *E. coli* and 1.5 μg/mL for *S. aureus*, were able to eradicate the biofilms.

## 4. Mechanism of Action of Ag NPs and Au NPs Loaded into Hydrogels

### 4.1. Mechanism of Antibacterial Action

Ag NPs and Au NPs exhibit antibacterial action against different antimicrobial-resistant bacteria, including Gram-positive and Gram-negative bacteria. In addition, their low reactivity and low toxicity, compared to Au and Ag ions, presents them as a relevant therapeutic strategy for drug-resistant bacterial infections [152,153].

Due to their proven potential, the understanding of the mechanisms of bactericidal action become relevant. In general, Ag NPs and Au NPs act through the following mechanisms: (a) adhesion and alteration to the surface of the microbial membrane; (b) penetration into bacterial cells promoting the breakdown of biomolecules and other intracellular damage; (c) induction of cellular toxicity by the generation of ROS that promote oxidative stress within the cell; and (d) inhibition of intracellular signal transduction pathways [148,154,155,156], shown in Figure 2.

Ag NPs and Au NPs act at the cell wall or membrane, as they adhere to these structures by electrostatic interactions. NPs release their positively charged ions, generated by metal oxidation, to the negatively charged bacterial cell surface [157,158]. In addition to the possibility of this interaction, Ag NPs also have an affinity for sulfur proteins in the microbial cell wall. The adhesion or accumulation of these nanostructures promotes irreversible morphological changes in the structure of the cell wall and membrane [154,159].

In this sense, it is evident that these nanoparticles interfere with the integrity of the lipid bilayer by denaturation. This can cause cell lysis, which increases the cell membrane permeability, affecting the cell’s ability to regulate the transport of substances and causes loss or leakage of cellular contents, such as cytoplasm, proteins, ions and the cellular energy reservoir (i.e., adenosine triphosphate) [160,161]. Thus, Ag NPs and Au NPs increase the permeability of bacterial cell membranes allowing the entry of antibiotics combined with NPs to potentiate the antibacterial effects [162,163].

Ag NPs and Au NPs can penetrate cells through existing porins in the outer or cytoplasmic membrane, promoting changes in cellular activity through the binding of NPs to cellular structures. This includes ribosomes, leading to protein synthesis reduction in the cytoplasm as well as of biomolecules such as proteins, lipids, and DNA. Among the biomolecules that are altered by the binding of Ag and Au ions, proteins and bacterial DNA are the most important [155,156,164].

Ag and Au ions released into the environment will bind to negatively charged protein, altering the protein structure, denaturing it, and interfering with the normal growth and metabolism of bacterial cells [164,165]. In addition, Ag ions bind to DNA via bonds with the sulfur and phosphorus components of the nucleic acid, causing denaturation, problems in DNA replication and stopping cell growth [148]. Au NPs, on the other hand, neutralize the plasmid charge and prevent its movement. These NPs can decrease the stability of the DNA structure by electrostatic repulsion [158,166].

Another mechanism of action of Ag NPs and Au NPs is the production of ROS and free radicals. These radicals promote oxidative stress in bacteria, inducing lipid damage, leakage of cellular biomolecules, protein aggregation, DNA destruction, and eventually, lead to cellular apoptosis. In addition, these NPs are considered the main agents for cell membrane disruption and DNA modification [148,167]. The production of ROS is normally dependent on the concentration of the nanostructures. ROS are generated after the uptake of free Ag and Au ions in the cells, which can alter the respiratory chain in the inner membrane by interacting with thiol groups forming Au-thiol groups. Ag-thiol groups promote the coagulation of respiratory enzymes, interrupting the production of adenosine triphosphate by altering the electron transport systems, and activating the apoptosis pathway [148,156].

Ag NPs and Au NPs can also alter bacterial signaling pathways. The mechanism of the signaling depends on the phosphorylation and dephosphorylation of cascade proteins or enzymes that are essential for cell activity and bacterial growth [164,165,168]. Due to the unique physicochemical properties of NPs, there is a possibility that these nanoparticles act as modulators of signal transduction in microbial cells. They may mediate bacterial cell apoptosis by disrupting the bacterial actin cytoskeletal network causing morphological changes in the bacterial form. Thus, bacterial cell membranes become more fluid, followed by cell rupture [155,169].

Au NPs have other different antipathetic activity mechanisms. Near-infrared radiation can be used to induce Au NPs to convert light into heat, which destroys the cytomembrane structure, and kills bacteria through lysis and disintegration by local heating. This therapy significantly reduces the number of bacteria, even at low concentrations, and includes specific mechanisms such as protein denaturation, cell fluid evaporation, cell structure breakdown, and blister formation. All these mechanisms damage the bacterial cell wall and promote cell wall penetration [157,160].

The dissolution state, the size and shape of the NPs in the exposure medium affect the release of ions and their antibacterial effect and mechanisms. Dissolution efficiency depends on synthesis and processing factors, as well as on the intrinsic characteristics of Ag NPs and the surrounding media [163,170]. Thus, studies claim that smaller NPs with a spherical or near-spherical shape are more prone to release silver. Therefore, reducing particle size and increasing the dispersibility can help to improve the antibacterial properties of the NPs. Additionally, this may facilitate adsorption, and penetration due to the greater surface area, and better dissolution in more acidic environments [152,162]. Thus, it is evident that the different mechanisms of action presented by the NPs increase the effect of their antibacterial action.

### 4.2. Mechanism of Action for Inhibiting Biofilm Formation

Regarding the literature, it is clear that these NPs are promising agents in inhibiting biofilm formation. However, the mechanism of action for inhibiting biofilm formation and the interrelation between Ag and Au NPs with biofilms is not completely understood. Some therapeutic targets are predicted to inhibit biofilm formation, such as: (a) the EPS network that facilitates the initial attachment of bacteria to the surface and increases bacterial resistance to host immunity and antibiotics; (b) the flagella, crucial structures for the initial communication between cells and the surface; (c) adhesion proteins, which allow for the initial attachment; and (d) quorum sensing (QS), bacterial cell–cell communication, in which bacteria give feedback via extracellular, signaling molecules to manage microbial virulence and to release autoinducers that increase in concentration as a function of bacterial cell density, shown in Figure 3 [168,171,172].

Cells treated with NPs may show alterations in their morphology, presenting a crumpled surface morphology, relatively elongated size, and no clear septum. These findings suggest that Ag NPs prevent bacterial cell division, causing membrane destruction, and preventing biofilm formation. With this effect, it can be seen that NPs induce morphological alterations reducing biofilm formation [159,161,173].

The inhibition of biofilm production also occurs by the generation of ROS induced by Ag NPs and by the release of silver. These mechanisms may inhibit the expression of genes related to motility and biofilm formation [156,167]. In a similar way, Au NPs cause mechanical damage to the cell wall through electrostatic interactions. Additionally, Au NPs can stimulate the production of ROS, and damage cellular structures, functions and proteins due to the release of metal ions [164].

Due to the positive correlation between EPS secretion and biofilm formation, EPS inhibition is also considered an alternative target to mitigate the biofilms of pathogenic bacteria [153,165]. For example, Ag NPs and Au NPs inhibit alginate production in a concentration-dependent manner. Alginate is a vital constituent of the EPS matrix. It helps bacteria attach to surfaces, protecting them from the host’s immune response, and thus making them resistant to antimicrobials. In this sense, one of the focuses of NP activity is to prevent the production and secretion of EPS matrix components, such as polysaccharides, proteins, and extracellular DNA or eDNA. These components, confer integrity to the biofilm and functional architecture, as well as resistance against antibiotics [171,172].

A fundamental step in biofilm formation is bacterial adhesion to a surface. In this sense, adhesion proteins, fimbriae, and flagella-mediated motility, that regulate the initial attachment of bacteria to a wide range of surfaces are therapeutic targets where NPs enact their antibiofilm activity [154,158]. Thus, some metallic nanoparticles can reduce bacterial adhesion on surfaces through the release of ions, inhibiting the enzymatic activity of proteins involved in peptidoglycan synthesis, delaying biofilm formation [156,165].

In addition to adhesion, another essential process is quorum sensing (QS). Some studies have indicated that metallic NPs can disrupt the production of QS molecules, especially the autoinducer (AI-2). Thus, the production of exopolysaccharides and rhamnolipids, motility, and some virulence factors necessary for QS-regulated biofilm production is substantially altered [171,172,174]. Thus, Ag NPs and Au NPs are considered very promising agents as a therapeutic strategy for coating surfaces and hospital utensils to prevent multidrug-resistant bacterial infections with biofilm production.

### 4.3. Mechanism for Biofilm Eradication

Most antibacterial agents have difficulty in penetrating the EPS matrix produced in a biofilm, making it difficult to eradicate and treat infections. Thus, it is essential to develop therapeutic strategies to solve this inconvenience. In this sense, the use of Ag NPs and Au NPs with intrinsic antimicrobial potential can act as biofilm-targeting agents, promoting its eradication [170,175]. Some studies have shown the mechanisms of action of these nanoparticles in biofilm eradication (Figure 4).

The level of biofilm–nanoparticle interaction depends on the physicochemical properties of the EPS, nanoparticles, and the environment around the biofilm. Thus, it is necessary to understand the NP transport process that includes movement to the biofilm–fluid interface, attachment to the surface of the biofilm and migration within the biofilm [176,177].

Initially, the penetration of nanoparticles into the EPS matrix is influenced by NP size, and by the interactions with components of the extracellular polymeric matrix with the NP surface properties, such as charge and functional groups [174,176]. Bacterial biofilms, in general, have a polyanionic and negatively charged matrix, which enables them to interact with positively charged Ag and Au metallic ions. Other, physicochemical-modulated characteristics are also important, especially electrostatic ones, such as the zeta potential [176,178,179].

When NPs begin to come into contact with an environment containing organic molecules, a corona-like coating is formed on the surface of the NPs. The nature of this corona influences NP–biofilm interactions [177,180]. After the initial contact, the nanoparticles begin to interact with macromolecules present in the biofilm, changing their known surface properties, related to size, strength, functionalization, and other biological properties [170,177].

Penetration, diffusion and antibiofilm effectiveness depend on physicochemical characteristics, such as adequate size, polydispersity index, purity, and zeta potential. The composition and structure of the biofilm are also important, including pore size, presence of water channels, hydrophobicity of the environment, and the chemical gradient of the matrix, as well as the ionic composition and concentration of the nanoparticle solution [178,180].

In this sense, NPs of 5–500 nm can penetrate into the biofilm water channels, affecting EPS matrix diffusion as a result of surface functional groups or charge interactions. NPs interact with bacterial cells through penetration and intracellular accumulation promoting the inhibition of protein function, DNA damage, translation disorders and/or transcriptional dysregulation. The pathogen viability is reduced by altering bacterial cell wall permeability (Figure 4) [164,166,169,173].

Studies have shown that NPs can bind to the negatively charged bacterial surface and the biofilm interacts electrostatically, promoting changes in quorum sensing and influencing bacterial growth (Figure 4). The results also showed a dose-dependent decrease in exopolysaccharide production, preventing the biofilm from maintaining its standard architecture (Figure 4) [169,171,172,174]. Biofilms treated with the formulation of Ag NPs and Au NPs may exhibit dispersed cell aggregates with acute structural destruction. Disruption of cells through membrane bubble whites is attributed to the close interaction of NPs and Ag and Au ions with the bacterial membrane, in addition to the possible ROS formation [156,167].

Some studies suggest that the main mechanism of biofilm destruction occurs through the binding of Ag NPs and Au NPs to the exopolysaccharide matrix. The biofilm structure is disrupted by recognizing the peptidoglycan structure present in bacterial membranes, causing physical damage, ion release, production of ROS, leading to oxidative stress, and DNA damage [154,164,167,169].

When bacteria are treated with Ag NPs morphological changes are revealed in the biofilm architecture. The irregular cell surface, suggesting cell lysis, relevant morphological damage to the cell wall, damage to membrane corrugation, changes in membrane polarization and/or permeability, and the distinct formation of an EPS matrix around bacteria are observed. In addition, electrostatic interactions between NPs and bacterial membranes cause them to disrupt, so that Ag NPs can penetrate the mature biofilm [156,164,173,174].

In addition, NPs can interfere with the condensation of the condensed cytoplasmic membrane, the pathways of bacterial metabolism, and the production of extracellular polysaccharides, leading to a change in the layout of the biofilm [170,180,181]. Due to these antibiofilm properties, Ag NPs and Au NPs are considered very promising for the treatment of multidrug-resistant bacterial infections and biofilm production [153,179,182].

## 5. Conclusions

Bacterial infections are a worldwide public health problem, and it has become more urgent due to some bacteria developing mechanisms of resistance to current antibiotics. This is a challenging problem, even more when microorganisms are capable of producing biofilms. However, this global public health challenge has motivated new research to develop therapeutic strategy and antibacterial biomaterials described in this review, such as hydrogels incorporating metallic NPs, in particular Ag and Au.

There is a wide variety of biocompatible polymers used in the production of hydrogels, such as alginate, chitosan, gelatin, konjac glucomannan, carbopol, carboxymethyl cellulose, carboxymethyl chitosan, poly-vinyl alcohol, gelatin methacrylate, polyacrylamide, and polyvinylpyrrolidone. These polymers have been shown to be excellent candidates as carriers of antibacterial NPs for the prevention and treatment of localized bacterial infections. Among those polymers, chitosan, Pluronic F127, gelatin, and poloxamer 407 seem to be the most promising. Some of these systems have achieved an antibacterial efficiency of 99.86%, 99.94%, 99.5%, and 99.0% against *E. coli*, *S. aureus*, *K. pneumoniae*, and *P. aeruginosa*, respectively. These bacteria are the most studied pathogenic microorganisms due to their pathogenic potential, ability to produce biofilms and to acquired antibiotic resistance. In vivo clinical trials, performed on animals, have shown that antibacterial hydrogels also help with healing and re-epithelialization.

To enhance the antibacterial action and inhibit the development of new resistance mechanisms by bacteria, new strategies have been explored, such as: (i) the surface functionalization of nanoparticles with other antimicrobial agents, such as polyethyleneimine, 2-mercapto-1-methylimidazole, 6-aminopenicillins acid, poly(allylamine hydrochloride), poly(ethylene glycol), and quercetin; (ii) drug encapsulation within the nanoparticles; (iii) incorporation of two metallic nanoparticles into the hydrogel; (iv) in situ photosynthesis of NPs into the hydrogel structure; and (v) photo-irradiation of the NP/hydrogel to eradicate bacteria. Therefore, there are many possible ways to fight against multidrug-resistant bacteria with the use of metallic nanoparticles incorporated into hydrogels.

Furthermore, the dispersion of metallic nanoparticles within the matrix of hydrogels significantly improves their ability to prevent and eliminate biofilm formation. This strategy was shown to be highly effective in the eradication of biofilms formed in wounds. Additionally, other techniques can be used alongside hydrogels to amplify their efficiency, such as phototherapy or photothermal therapy. Thus, these hydrogels are promising antibiofilm agents and may be critical in treating bacterial infections associated with biofilms in the future.

Although there are many results in the literature and examples of in vitro and in vivo studies in animals, clinical applications of biopolymers are limited to a few reports. Examples of clinical applications in human patients are even scarcer for Ag NPs and Au NPs incorporated into biopolymers. Real case applications are essential to demonstrate their technical, economical and clinical feasibility. Therefore, great opportunities for the development of such composites are on the horizon.

## Figures and Tables

**Figure 1 antibiotics-12-00104-f001:**
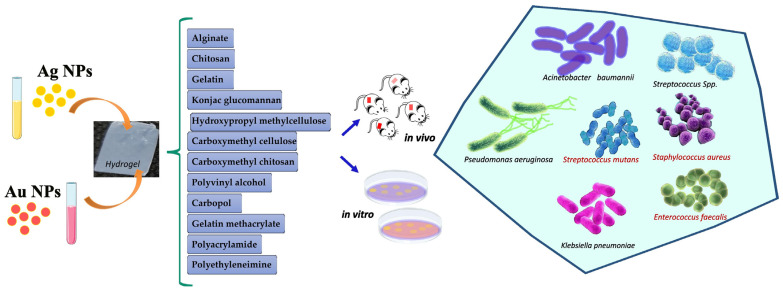
Silver and gold NPs loaded into hydrogels for antibacterial application.

**Figure 2 antibiotics-12-00104-f002:**
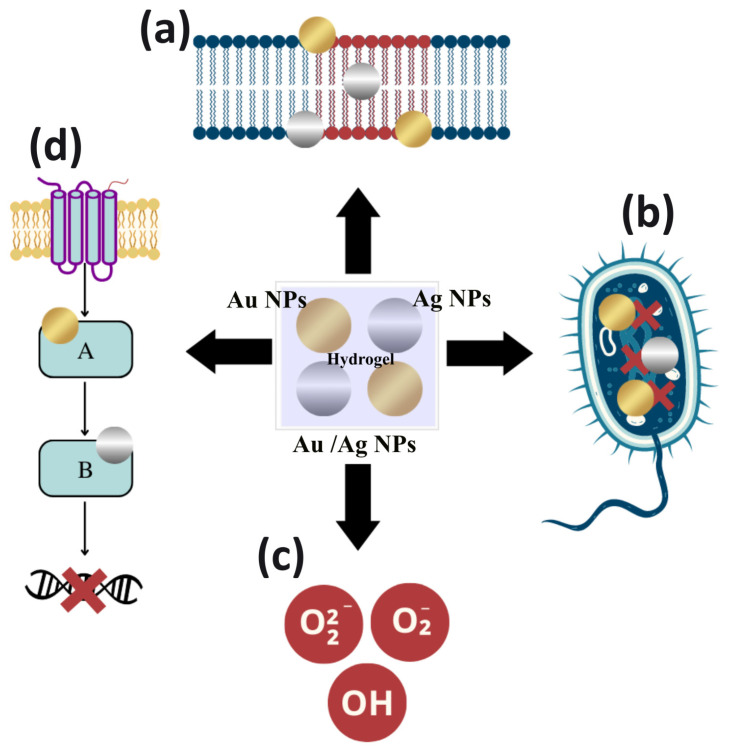
The mechanisms of bactericidal action of Au NPs and Ag NPs loaded into hydrogels.

**Figure 3 antibiotics-12-00104-f003:**
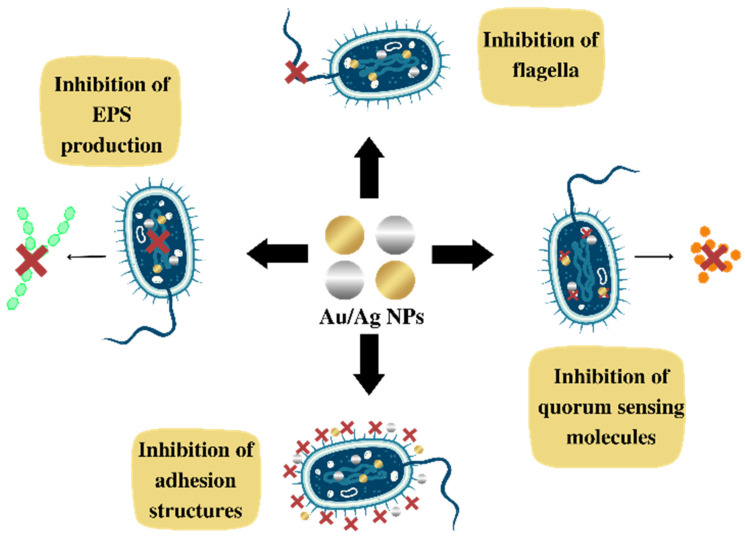
Mechanism of inhibiting biofilm formation by Au NPs and Ag NPs.

**Figure 4 antibiotics-12-00104-f004:**
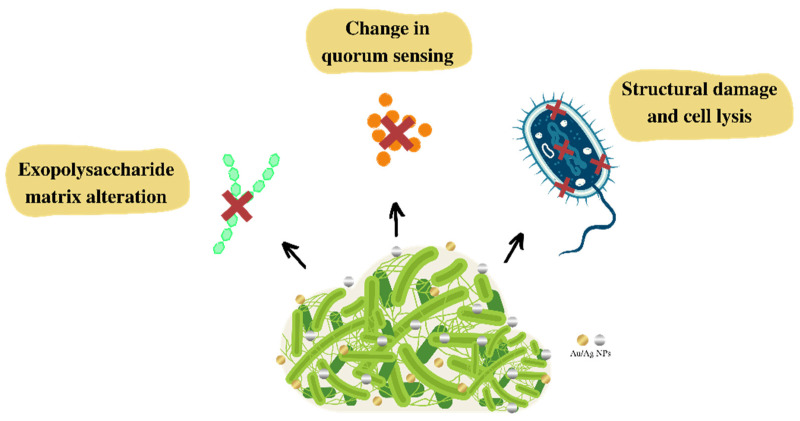
Mechanism of biofilm eradication of Au NPs and Ag NPs.

**Table 2 antibiotics-12-00104-t002:** Ag NPs loaded into hydrogel for antibacterial application.

System	Materials	Ag NP Properties (Size and Surface Charge)	NP Synthesis Method	Bacteria	Target	Antibacterial Properties: Inhibition Zone (mm) and MIC Values	Ref.
Ag–ODex HA-ADH/HACC	Dextran, sodium hyaluronic, chitosan quaternary ammonium salt, and AgNO_3_	50–190 nm	Chemical reduction, in situ, Schiff-base reaction to form hydrogel	*E. coli* ATCC8739, *S. aureus* ATCC14458, and *P. aeruginosa* CMCCB10104	In vitro; In vivo, rats	The Kirby–Bauer (KB) method. The inhibition zone was 24, 24, and 27 mm, respectively	[27]
Ag/CS	LiOH, KOH, CH_4_N_2_O, AgNO_3_, and Na_3_C_6_H_5_O_7_	Spherical and ellipsoidal NPs; 4.45–9.22 nm	Chemical reduction with sodium citrate, in situ	*E. coli* and *S. aureus*	In vivo; rats	Antibacterial activity: 99.86% and 99.94%, respectively	[22]
Ag/CM- βCD	Chitosan, NaBH_4_, AgNO_3_, NaOH, cyclodextrin, CH₃CO_2_H, and C_5_H_8_O	50 nm	Chemical reduction with NaBH_4_, in situ	*E. coli* and *S. aureus*	In vitro	The inhibition zone increased when the CM-βCD concentration was increased in the hydrogel	[25]
Ag/N, O-carboxymehtyl chitosan (N, O-CMC)	Chitosan, AgNO_3_, C_10_H_16_N_2_O_8_ (EDTA), CaCl_2_, FeCl_3_, and C_2_H_3_ClO_2_	25 nm	Chemical reduction using C_2_H_3_ClO_2_	*E. coli* ATCC25922, *S. aureus* ATCC35556, MRSA ATCC 43300, *P. aeruginosa* ATCC47085, and *K. pneumonia* ATCC700603	In vitro, L929 cells	MIC values: 48.5 mg/mL for *P. aeruginosa;* 32.0 mg/mL for *S. aureus* and MRSA; 17.5 mg/mL for *E. coli*, and 23.0 mg/mL for *K. pneumonia*	[37]
Ag/OKGM-CMCS	Oxidized konjac glucomannan (OKGM) and Carboxymethyl chitosan (CMCS)	60 nm	Schiff-base reaction	*S. aureus* and *E. coli*	In vitro, L929 cells; In vivo, rats	The Ag/hydrogel achieved high antimicrobial activity, but the inhibition zone values were not displayed	[23]
Ag/KGM	Eggs, konjac glucomannan, AgNO_3_, and NaOH	9.5–30.2 nm	In situ	*S. aureus* and *E. coli*	In vitro; In L929 cells; in vivo, rabbits	Good antibacterial efficiency on rabbits’ skin infections	[111]
Ag/CMC/PVA/EGDE	Carboxymethyl cellulose (CMC), polyvinyl alcohol (PVA), and ethylene glycol diglycidyl ether (EGDE)	8–14 nm	Microwave radiation	*E. coli*, *K. pneumoniae*, *P. aeruginosa*, *Proteus vulgaris*, *S. aureus*, and *Proteus mirabilis*	In vitro, patient urine	The inhibition zone: 16.6 mm for *E. coli*, 15.8 mm for *K. pneumoniae*, 15.6 mm for *P. aeruginosa* and 15.2 mm for *P. vulgaris*	[13]
QCT-Ag/Carbopol- *aloe vera*	Carbopol 934, AgNO_3_, QCT, polyvinylpyrrolidone (PVP), *Aloe vera*, C_3_H_8_O_3_, and NaBH_4_	44.1 nm; ζ: −14.76 mV	Chemical reduction with NaBH_4_	*S. aureus* MTCC 3160 and *E. coli* BL-21	In vitro, L929 cells; In vivo, mice skin	The inhibition zone: 17 mm for *E. coli* and 19 mm for *S. aureus*	[63]
Ag/graphene	AgNO_3_, C_7_H_10_N_2_O_2_, (NH_4_)_2_S_2_O_8_, and NH_3_H_2_O	39 nm	Hummer’s method	*E. coli* and *S. aureus*	In vitro, L929 cells; In vivo, rats	The disc diffusion method. Large Ag concentration led to great antibacterial activity using 5:1% wt. of Graphene	[41]
Ag/poly(vinyl alcohol)/chitosan/graphene	Graphene, chitosan, CH_3_CO_2_H, KNO_3_, AgNO_3_, and K_2_HPO_4_	6.38–10.00 nm	Electrochemical synthesis in situ using 90 V	*E. coli* ATCC 25922 and *S. aureus* TL	In vitro, MRC-5 and L929 cells;	The inhibition zone: 15.5 mm for *S. aureus* and 13.5 mm for *E. coli*; great antimicrobial activity with the 0.25Ag/PVA/0.5CHI/Gr	[70,71,72]
Ag/PEI- graphene oxide	Pluronic F 127, graphene oxide, C_8_H_17_N_3_.HCl, AgNO_3,_ NH_4_OH, and NaCl	10 nm; ζ: 42.6 mV	Amidation reaction with Ag(NH_3_)_2_OH by microwave reactor	*E. coli* and *C. albicans*	In vitro	*E. coli* (99.86%) and *C. albicans* (99.94%)	[66]
Ag/PAA-MBA	K_2_S_2_O_8_, NaBH_4_, PVP, C_3_H_5_NO, C_6_H_9_Na_3_O_9_, and AgNO_3_	Spherical: 12.7 nm; triangular: 37.1 nm; hexagonal: 26.9 nm	Chemical reduction using NaBH_4_	*E. coli* W3110	In vitro	The spherical and triangular shapes of the Ag NPs displayed better antibacterial activity than the rod-shaped NPs.	[40]
Ag/halloysite/gelatin methacrylate	AgNO_3_, NaBH_4_, (CH_3_)_2_SO, and C_2_H_4_O	Ag NPs changed the microstructure and roughness of the hydrogel	In situ by photopolymerization using UV radiation (365 nm and 400 W)	*E. coli* ATCC 8739 and *S. aureus* ATCC 29213	In vitro; In vivo, crania of rats	The inhibition zone test showed that the hydrogel restrained the growth of the bacteria	[65]
Ag/KGM	Chitosan, carboxymethyl, β-cyclodextrin, etc.	50 nm	Chemical reduction	*S. aureus* and *E. coli*	In vitro	Inhibition zone: 22 and 19 mm, respectively	[25]

**Table 3 antibiotics-12-00104-t003:** Au NPs loaded into hydrogel for antibacterial application.

System	Materials	Au NP Properties (Size and Surface Charge)	NP Synthesis Method	Bacteria	Target	Antibacterial Properties: Inhibition Zone (mm) and MIC Values	Ref.
AuC/liposome	Cationic phospholipid liposomes, acrylamide, (glycol) dimethacrylate (PEGDMA)	97.1 nm, ζ: −25.3 mV	Chemical reduction with NaBH_4_	*S. aureus* MRSA252	In vitro; in vivo, mice	No skin reaction after 7-day treatment. Hydrogel activity was influenced by pH	[133]
Au NSt/alginate	Sodium alginate (SA), CaCl_2_, and polyethylene imine (PEI)	Core diameter: 25 nm; Spikes size: 50 nm, 70 nm, and 120 nm	Chemical reduction with trisodium citrate	*S. aureus* MTCC1430 *P. aeruginosa* MTCC 1934 *E. coli* MTCC 443	In vitro, NIH-3T3; in vivo, rats	The plate count method; the antimicrobial activity: 35.4% (*S. aureus*), and >80% (*P. aeruginosa* and *E. coli*.)	[126]
Au/poly (acrylamide-co-alginate)	Acrylamide (AM), alginate (SA), N,N-methylenebisacrylamide, and HAuCl_4_	8 nm	In situ, chemical reduction	*E. coli*	in vitro	Optical absorbance around 0.05–0.75. *E. coli* did not growth more after 2 h 30 min	[59]
CS-Au–MMT/gelatin	2-mercapto-1-methylimidazole (MMT), tannin acid, chitosan (CS), and gelatin	10.07 ± 2.34 nm 8.32 ± 1.97 nm	Chemical reduction	*S. aureus* ATCC 25923, *E. coli* ATCC 25922 MRSA	In vitro, L929 and L02; in vivo, rabbits	In situ; the microtiter broth dilution method, and MIC < 20 µM for all bacteria	[60]
Au-Ag/CS/TEOS	HAuCl_4_, HNO_3_, chitosan, and tetraethyl orthosilicate (TEOS)	Ag: 16 ± 25% nm Au: 19 ± 18% nm	Polymerization reaction and drop casting method	*E. coli*	In vitro	Crystal violet attachment; 80% inhibition of *E. coli* on the surface	[136]
Au–APA/gelatin	6-aminopenicillanic acid (APA), gelatin, and HAuCl_4_,	5 nm	Chemical reduction by NaBH_4_	*E. coli*, *K. pneumoniae*, *P. aeruginosa*, MDR *E. coli*, and MDR *K. pneumoniae*	In vivo, rats	The microtiter broth dilution method; MIC were 2.5 µg/mL against *E. coli* and *K. pneumoniae*, >5 µg/mL against *P. aeruginosa*, 5 µg/mL against MDR *E. coli* and MDR *K. pneumoniae*	[61]
Au/HPMC	Tetrachloroauric acid, cetyltrimethyl ammonium bromide, ascorbic acid, NaBH_4_, AgNO_3_, and hydroxypropyl methylcellulose (HPMC)	82.5 nm; ζ: 34.8 mV	Chemical reduction method using CTAB and NaBH_4_. Au NPs were embedded into HPMC	*Staph. aureus* ATCC 10400, *E. coli* ATCC 25922, and *C. albicans* ATCC 90028	In vitro; in vivo, rats	Micro broth dilution assay. MIC and MBC: 0.25 and 0.1 nM/mL for *Staph. aureus*, MIC and MBC: 0.125 and 0.125 nM/mL for *E. coli*, and MIC and MBC: 0.25 and 0.5 nM/mL for *C. albicans*,	[24]
Au/Silk	HAuCl_4_, sodium citrate, bombyx mori cocoons, NaCO_3_, and LiBr	13 nm	Chemical reduction using sodium citrate	*E. coli* ATCC 25922 *and S. aureus* ATCC 25923	In vitro; in vivo, mice	Killed 80% of bacteria in 10 min; using a laser exposure time of 15 min and 600 mW, the zone inhibition was about 16 mm^2^	[26]
Au/CA-DEG-IAA	Citric acid (CA), diethylene glycol (DEG), and indolylacetic acid (IAA)	17 nm	In situ; chemical reduction with Na_3_C_6_H_5_O_7_	*S. aureus*	In vitro	The diffusion method; Inhibition zone: 8.33–11.6 mm	[140]
Au/CA-DEG-IAA Ag/CA-DEG-IAA	Citric acid (CA), diethyleneglycol (DEG), and indole-3-acetic acid (IAA)	Au NPs: 8–30 nm Ag NPs: 4–12 nm	Condensation polycondensation; chemical reduction with Na_3_C_6_H_5_O_7_	*S. aureus*, *E. coli*, and *Bacillus cereus*	In vitro	Inhibition zone (mm): 25 and 15 mm, 23 and 14 mm, 25 and 15 mm	[128]
Au/poloxamer 407	CTAB (C_16_H_33_N(CH_3_)3Br), PAA (polyacrylic acid), PAH (poly(allylamine hydrochloride), and PEG (Poly(ethylene glycol)	Rod shape: 49.2 nm Spherical shape: 29.2 nm	Chemical reduction using Na_3_C_6_H_5_O_7_	*S. aureus ATCC 29213*, and *P. aeruginosa* ATCC 27853	In vitro; in vivo, rats	Reduction in bacterial viable count was >99.5% and 99.0> against *S. aureus* and *P. aeruginosa* using PAH-Au NPs and PEG-Au NPs.	[130]
FPAu	Polyethyleneimine (PEI), Polydopamine (PDA), Pluronic F127, 4-hydroxy benzaldehyde (PHBA), HAuCl_4_, and K_2_CO_3_	10 nm	Chemical reduction with NaBH_4_, and polyvinyl pyrrolidone (PVP)	*E. coli* *S. aureus*	In vitro; in vivo, rats	The plate count method; inhibited bacterial growth in 75% after 2 h	[68]
Au–PDA/PNAGA	HAuCl_4_, NaBH_4_, dopamine hydrochloride, and N-acryloyl glycinamide (PNAGA)	Diameter 32 nm and length 54 nm	Seeded growth method Polymerization	*S. aureus* ATCC29213 and *E. coli* ATCC25922	In vitro, L929 cells; in vivo, rats	97.6%; 98.4%	[131]
Ag–Au/carbopol	Carbopol^®^ 980, acrylamide, AgNO_3,_ and HAuCl_4_	2–8 nm	In situ reduction using mint leaf extract	*Bacillus* *E. coli*	In vitro	The disc method; inhibition zone: 18.5 mm 18.1 mm	[64]
Ag–Au/CMT	AgNO_3_, KAuCl_4_, and carboxy methyl tamarind (CMT)	187 nm	Seeded growth method	Clinical *E. cloacae isolate* Ec18, *E. cloacae* BAA-1143, ATCC, and *E. coli* BAA-2469, ATCC	In vitro; in vivo, mice	The disc method. MIC: 6 µg/mL, 6 µg/mL, and 3 µg/mL	[29]
Au/Ag–gelatin	glutathione (GSH), HAuCl_4_, AgNO_3_, and N-hydroxysuccinimide (NHS)	Au NCs: 1.5–3.5 nm Au/Ag: 102 nm	Au/Ag NCs was incorporated into gelatin after NPs synthesis	*P. aeruginosa*	In vitro, pigskin	Inhibition zone: 31.9 mm	[144]
Au or Ag/silk fibroin	AgNO_3_, HAuCl_4_, and cocoons of *Bombyx mori* silkworm	Au NPs: 9–55 nm Ag NPs: 12–69 nm	In situ chemical reduction	*S. aureus* ATCC 33591, MRSA and *P. aeruginosa* ATCC 27853 *S. aureus* ATCC 25923, MSSA and *E. coli* ATCC 25922 *S. epidermidis* RP62A ATCC 35984.	In vitro, MG63 cells	Using sessile and planktonic bacteria. 0.1% of Au NPs were effective against *S. aureus*, and *E. coli* while 0.5% of Au NPs was antibacterial against *P. aeruginosa*	[28]
Au–ZIF8/OSA-GelMA	HAuCl, Na_3_C_6_H_5_O_7_, polyvinyl pyrrolidone (PVP), gelatin, Zn(NO_3_)_2_.6H_2_O, CH_6_N_4_O, oxidized sodium alginate (OSA), and carbohydrazide-modified methacrylated gelatin (GelMA-CDH)	15 nm; ζ: −4.8 mV	Chemical reduction with Na_3_C_6_H_5_O_7_; Schiff-base reaction, and radical polymerization	*E. coli* ATCC 25922 *S. aureus* ATCC 29213	In vitro, NIH-3 T3 cells	The number of bacteria colonies decreased by more than 99%	[69]
Au/C/PAM	Acrylamide monomer, cellulose, HAuCl_4_, and ciprofloxacin	Length: 5 µm and diameter 70 nm	In situ; chemical reduction with Na_3_C_6_H_5_O_7_	*E. coli*, *S. flexneri*, *Bacillus cereus*, and *Listeria inuaba*	In vitro, L929 cells	The diffusion method; the antibacterial activity was 95% against the *E. coli*, and 79% against the *S. flexneri*.	[146]

## Data Availability

Not applicable.

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
