# Peer review of "Advanced Hydrogels Combined with Silver and Gold Nanoparticles against Antimicrobial Resistance"

_antibiotics, 2023, doi:10.3390/antibiotics12010104_

Round 1
Reviewer 1 Report
In the manuscript entitled “Advanced Hydrogels Combined with Metallic Nanoparticles Against Antimicrobial Resistance”, the authors describe the production and characterization of new nanoformulations, with important antimicrobial properties.
In introduction, the novelty and the purpose of this work are well defined. A series of specific studies were exposed and the corresponding results are presented in a concise manner. However, in the attached document you can find some points that I consider important to complete / modify!

Author Response
Dear Reviewer,
Please see the attachment.
Prof. Dr. Sotiris K Hadjikakou
Dr. Christina N. Banti
Guest Editors of Antibiotics
Ms. Thalia Zhang
Assistant Editor, MDPI
Vitória de Santo Antão, December 21st 2022.
Dear Editors and Referees,
Thanks for your message with the report of the Referees of our review entitled Advanced Hydrogels Combined with Silver and Gold Nanoparticles Against Antimicrobial Resistance by Moreno et al., to be submitted to the special issue of MDPI Antibiotics entitled “Silver and Gold Compounds as Antibiotics”.
We have carefully taken into account all the comments and suggestions made by them. We agree with all of them and we sincerely believe that they improved the quality of our work. We explain below all the changes which were done, as well as the place in the manuscript they were inserted, according to each comment. Please check the following our reply (blue font) of each reviewer’s comments (black and italic font). For the sake of clarity, a marked “Moreno Bold” version is also included.
Reviewer #1: In the manuscript entitled “Advanced Hydrogels Combined with Metallic Nanoparticles Against Antimicrobial Resistance”, the authors describe the production and characterization of new nanoformulations, with important antimicrobial properties.
In introduction, the novelty and the purpose of this work are well defined.
A series of specific studies were exposed and the corresponding results are presented in a concise manner.
However, here are some points that I consider important to complete / modify:
Major changes:
- Revise the wording of the paragraphs! Link the ideas inside a paragraph, make connections. Sometimes they seem like strings of ideas, without connections between them! Some phrases are very difficult to understand! Please revise the English language!
During the revision, I will mark the phrases that especially require major improvements!
We agree with the Reviewer #1. The English language of the review has been carefully proofread and all alterations done are marked in bold font in the manuscript. The suggestions were gratefully accepted. We believe the manuscript has been considerably improved.
Minor changes:
- In many places, punctuation marks are missing (especially commas), which makes it difficult to read and understand some ideas.
We agree with the Reviewer #1. Punctuation marks such as commas and points were inserted in the manuscript in several lines, as for example in lines: 33, 65, 149, 192, 211–212, etc.
- Reword lines 63-64. …”one can highlight S.aureus is resistant”???
The numbering of the lines in this new version of the manuscript has changed.
As suggest by Reviewer #1, lines 64-65 have been rewritten as “…, these bacteria can be highlighted: S. aureus, which is resistant to methicillin and vancomycin” on page 2.
- Reformulate the phrase from lines 147-149, for a better understanding of the idea.
Done. Lines 148-151 have been rewritten as “The effect of Ag NPs on the bacterial membrane is related to their physicochemical properties, such as size, shape, surface area, surface charge, oxidation state, and surface chemistry. It has been reported that Ag NPs with small size and colloidal stability are preferred rather than those susceptible to aggregation”.
- Line 182: Ag nanoclusters are effective insetead of be effective.
Thanks for suggestion. Line 184 has been rewritten as “Ag nanoclusters (NCs) are effective for this type of application”.
- Line 183: NCs are NPs whose sizes instead of which sizes.
As suggest by Reviewer #1, line 185 has been revised as “NCs are NPs whose sizes are smaller than 2 nm”.
- Line 198: „nanostructures a and”, please delete the „a”.
As suggest by Reviewer #1, line 201 has been rewritten as “size of the nanostructures and participate”.
- Line 206: „regeneration” instead of „generation”.
Done. Line 209 has been rewritten as “the healing and regeneration of the skin”.
- Line 209: „the second most abundant biopolymer in nature, after??? After cellulose?...please complete!
Done. Lines 213-214 have been rewritten as “the second most abundant biopolymer in nature, after cellulose. It has been”.
- Lines 220-221: I suggest you connect the ideas like this: „…., while recorded inhibition zones were...”
Thanks for the suggestion. Lines 224-225 have been revised as “The hydrogel displayed antibacterial properties against E. coli ATCC 8739, S. aureus ATCC 14458, and P. aeruginosa CMCCB10104, and the inhibition zones were 24, 24, and 27 mm, respectively”.
- Line 223: Please rephrase!
Done. Lines 226-228 have been rewritten as “These results were associated with the hydrogel's positive charge due to the quaternate chitosan's cationic group, that favor the interaction with the negative charge bacteria cell walls”.
- Lines 232-233: Please combine the sentences into a single idea.
Done. Lines 237-238 have been rewritten as “The wound contraction was 70.5% on 4th day and after it was 99.75% on 14th day”.
- Line 235: ”from cell membrane, and impair the respiratory function of bacteria”
Done. Lines 240-241 have been rewritten as “from bacterial cell membrane. The Ag NPs would merge with DNA´s bacteria damaging the bacterial replication and impair the respiratory function of bacteria”.
- Lines 319-320: Please rephrase!
Done. Lines 331-332 have been rewritten as “The antimicrobial activity against E. coli was 99.86%, and 99.94% against C. albicans, using 10 µg/mL of hydrogel”.
- Line 322: „bacterial cell wall” instead of „bacterial wall cells”
Done. Line 334: “bacterial wall cells” has been revised as “bacterial cell wall”.
- In table 2: You also have antibacterial properties expressed as MIC values! Please complete the table header.
Done. In table 2 and table 3: The table header “Antibacterial properties: Inhibition zone (mm)” has been rewritten as “ Antibacterial properties: Inhibition zone (mm) and MIC values”.
- Line 375: Please explain „loaded ls hydrogel”????
Done. Line 388: “Ag NPs-loaded ls hydrogel against B. subtilis” has been rewritten as “Ag NPs-loaded hydrogel against B. subtilis”.
- Line 413 the tested strain” instead of „the strain tested”.
Done. Line 426: “the strain tested” has been rewritten as “the tested strain”.
- Line 427: „2 MIC concentration”- what is the MIC unit in this case.
Done. Line 440: “the hydrogel formulation at 2 MIC concentration” has been revised as “the hydrogel formulation at 2 MIC (μg/mL) concentration”.
- Line 531: „the male” it is with lowercase letter.
Done. Line 544: “The Male Wistar rats were used” has been revised as “The male Wistar rats were used”.
- Line 532: The MICs- it is the plural form. Please replace "and" with "," (to the stringing of bacterial strains.
Done. Lines 544-546: “The MIC against S. aureus ATCC 10400 and E. coli ATCC 25922, and C. albicans ATCC 90028 was 0.125 – 0.25 ng/mL” have been revised as “The MICs against S. aureus ATCC 10400, E. coli ATCC 25922, and C. albicans ATCC 90028 were 0.125 – 0.25 ng/mL”.
- Line 533: The MBCs- it is the plural form.
Done. Line 546: “The minimum bactericidal concentration (MBC) was 0.1 – 0.5 ng/mL” have been revised as “The minimum bactericidal concentrations (MBCs) were 0.1 – 0.5 ng/mL”.
- Lines 555-557: Please rephrase to be more clear that is about the diameter of the inhibition zone.
Done. Lines 568-569: “The antibacterial performance against S. aureus was 8.33 – 11.67 mm using 1000 – 2000 µg/mL” have been revised as “The antibacterial performance against S. aureus showed an inhibition zone from 8.33 to 11.67 mm, using 1000 – 2000 µg/mL”.
- Line 561: „The inhibition zones…were…” instead of „the zone inhibition…was”
Done. Lines 574-575: “The zone inhibition of Au/hydrogel was 14 mm against E. coli, 16 mm for S. aureus, and 15 mm for B. cereus” have been rewritten as “The inhibition zones of Au/hydrogel were 14, 16, and 15 mm against E .coli, S. aureus, and B. cereus, respectively”.
- Line 577: please connect- „polymer which displays….”
Done. Lines 590-591: “Also, Pluronic F-127 is a synthetic thermoresponsive polymer. Pluronic displays sol-gel transition” have been revised as “Pluronic F-127 is a synthetic thermoresponsive polymer which displays sol-gel transition near 37 oC”.
- 25. Lines 581-587 and Lines 600-606: please connect the ideas together; please rephrase.
Done. Lines 594-599 have been revised as “This FPAu biomaterial was obtained by the double crosslinking of pluronic F127, 4-hydroxy benzaldehyde, K2CO3, and modified polydopamine NPs. The number of colony-forming decreased rapidly after 2 hours of contact between bacteria and hydrogel at in vitro tests. The antibacterial activity against S. aureus and E. coli was 87.5% and 83%, respectively. This study suggested that antibacterial property of FPAu hydrogel is due to branched polyethyleneimine linked on Au NPs surface”.
Lines 612-618 have been revised as “Some authors explored alternative methodologies, for example, by incorporating dual metallic nanoparticles, such as Ag and Au NPs, as core@shell nanoparticles into the porous structure of Carbopol-base hydrogel [64]. The particle size was 5 ± 3 nm and the Ag@Au NPs presented several morphologies into the hydrogel. The inhibition zones against B. cereus and E. coli were 18.5 and 18.1 mm, respectively. This study showed that Ag@Au NPs inhibit the growth of bacteria by forming pits between NPs and the cell wall. This interaction resulted in bacteria death due to leak of molecules and proteins from the wall surface”.
- Line 609: tests using mammalian cells are „in vitro”, not „in vivo” tests.
Done. Lines 621-622: “hydrogel against MDR E. coli, E. cloacae, and S. aureus MRSA and tested in vivo at mammalian cells” have been revised as “hydrogel against MDR E. coli, E. cloacae, and S. aureus MRSA for in vitro tests using mammalian cells”.
- Line 611: what is „ lag phase”??..... Ag NPs at a concentration…, while the MIC presented values were of 3 and 6…”…please complete!
Done. Lines 623-627 “The growth profile of cells exposed to hydrogel was a lag phase after 4 h in Au@Ag NPs concentration of 1 – 3 µg/mL for E. coli and 3-12 µg/mL for MRSA, while the MIC presented values were of 3 and 6 µg/mL for E. coli and MRSA, respectively” have been rewritten as “The growth profiles of cells were studied at different concentrations of Au@Ag NPs. E. coli shown an extended lag phase exposed to Au@Ag NPs at a concentration of 1 to 3 µg/mL, while at a concentration of 3 to 12 µg/mL for MRSA in presence of Au@Ag NPs. The lag phase is the earliest period of the bacterial growth cycle, which the bacteria adjust to environment, and cells increase in size [144]. The reported MICs values were 3 and 6 µg/mL for E. coli and MRSA, respectively”.
- 28. Line 620: „efficient bactericidal effect” instead of „efficient bacterial killing”.
Done. Lines 636-637: “efficient bacterial killing and makes it a promising material” have been revised as “efficient bactericidal effect and turns this hydrogel a promising material”.
- 29. Line 622: „assays” not „essays”
Done, Line 638: “zone essays’ has been rewritten as “zone assays”.
- Lines 625-627: Please rephrase and… complete ”higher than” ..than what? Please finish the idea!
Done. Lines 642-643: “the efficiency of Au/Ag@gelatin was higher than, with 31.9 mm of inhibition zone, against 25.1 mm for Au@gelatin” have been revised as “The inhibition zone of Au/Ag@gelatin was 31.9 mm higher than the Au@gelatin with 25.1 mm”.
- Lines 629-632: Please link and correlate the ideas.
Done. Lines 645-648 have been rewritten as “The size distribution of Au NPs was 9 to 55 nm and for Ag NPs was 12 to 69 nm. The NPs concentration influenced the antimicrobial activity. For example, the hydrogel loaded with a concentration > 0.5% of Au NPs was efficient against the S. aureus ATCC 33591, MRSA, and P. aeruginosa ATCC 27853”.
- Line 640: „Thus, improving the …” it looks like the end of a sentence, not a sentence as such! Please check!!!
Done. Lines 656-657 “Thus, improving the antimicrobial activity of Au NPs” have been revised as “Au NPs/ZIF-8 strategy that can improve the antimicrobial activity of gold NPs”.
- 33. Line 646: „using mice for an in vivo model” please correct.
Done. Line 663: “using mice as a model” has been revised as “using mice for an in vivo model”.
- Line 657: Shigella not Shugella; also the species will be written in lower case everywhere.
Done. Line 673 and Table 3: “Shugella Flexneri” has been revised as “S. flexneri”.
- Lines 676-692: This is an example of a paragraph where all the ideas are connected and flow naturally from one to the next.
Thanks for your comments.
- Lines 710-713: Rephrase please; the phrase is ambiguous.
Done. Lines 725-729 have been rewritten as “The authors found colony-forming unit (CFU) differences when comparing the Au NRs-treated biofilm (188.6 ± 26.7 CFU) with the control group (337.3 ± 2.82 CFU). The Au NRs successfully eliminated E. faecalis biofilm. The laser application time, however, was 20 minutes, which is longer than would be feasible for an in vivo scenario. The researchers suggest that”
- Line 724: I would say „while low concentrations….”.
Done. Lines 738-739: “Gram-negative bacteria and low concentrations, such as 1 μg/mL for E. coli and 1.5 μg/mL for S. aureus, were able to eradicate biofilms” have been revised as “Gram-negative bacteria. Low concentrations, such as 1 μg/mL for E. coli and 1.5 μg/mL for S. aureus, were able to eradicate biofilms.”
- Lines 730-733: Please rephrase!
Done. Lines 747-749: “Gram-negative, in addition to having low reactivity and toxicity compared to Au and Ag ions, configuring themselves as a therapeutic strategy for the treatment of resistant bacterial infections” have been rewritten as “Gram-negative. In addition, their low reactivity and low toxicity, when compared to Au and Ag ions, tuns them as a relevant therapeutic strategy for resistant bacterial infections treatment”.
- Line 736: I would say „…act through the following mechanisms”.
Done. Lines 751-752: “In general, it can be said that Ag NPs and Au NPs act through a) adhesion” have been revised as “In general, Ag NPs and Au NPs act through the following mechanisms: a) adhesion”.
- Line 767: Please rephrase the unnecessary repetition.
Done. Lines 786-787: “These NPs can decrease the stability of the DNA structure by electrostatic repulsion due to DNA” have been revised as “These NPs can decrease the stability of the DNA structure by electrostatic repulsion”.
- Lines 772-777. A long paragraph and a single sentence, hard to follow and understand! Please rephrase.
Done. Lines 792-798 have been rewritten as “The production of ROS is normally dependent on the concentration of the nanostructures. ROS are generated after the uptake of free Ag and Au ions in the cells, which can alter the respiratory chain in the inner membrane by interacting with thiol groups forming Au-thiol groups. Ag-thiol groups promote the coagulation of respiratory enzymes, interrupting the production of adenosine triphosphate by altering electron transport systems, and activating the apoptosis pathway”.
- Line 783: Separate the ideas: „This is possible by disrupting the bacterial…”
Done. Lines 803-806: “mediating bacterial cell apoptosis by disrupting the bacterial actin cytoskeletal network causing morphological changes in the bacterial form, thus increasing fluidity in the bacterial cell membrane that is followed by the rupture of the cells” have been revised as “They may mediate bacterial cell apoptosis by disrupting the bacterial actin cytoskeletal network causing morphological changes in the bacterial form. Thus, bacterial cell membrane become more fluid, which is followed by the rupture of the cells”.
- Lines 806-807: It's a meaningless phrase! Please check!
Done. Lines 827-828: “However, the mechanism of action for inhibiting biofilm formation and the interrelation between Ag and Au NPs with biofilms” have been revised as “However, the mechanism of action for inhibiting biofilm formation, and the interrelation between Ag and Au NPs with biofilms are not completely understood”.
- Line 824: I would say „promoting the cessation of the cellular structures functions”.
Done. Lines 846-847: “electrostatic interactions, as well as stimulate the production of ROS, promoting the cessation of the functions of cellular structures and proteins” have been rewritten as “electrostatic interactions. Also, Au NPs can stimulate the production of ROS, and damaging the cellular structures functions and proteins”.
- Line 847: „hospital ustensils”…please correct!
Done. Line 874: “hospital ustensils” has been revised as “hospital utensils”.
- 46. Line 923: „These polymers” instead of „this is polymers”
Done. Line 957: “This is polymers have” has been rewritten as “These polymers have”.
- Line 926: „some of these”…please correct!
Done. Line 960: “Some os these systems” has been revised as “Some of these systems”.
- Line 947: I would move „in the future” at the end of the sentence.
Done. Lines 980-981: “antibiofilm agents and may, in the future, be critical parts of treating bacterial infections associated with biofilm” have been revised as “antibiofilm agents and may be critical parts of treating bacterial infections associated with biofilm in the future”.
- Line 952: I would use: “economical”, in this context!
Done. Line 986: “technical, economic, and clinical feasibility” has been rewritten as “technical, economical, and clinical feasibility”.
We would like to thank you very much for reviewing our manuscript carefully and giving us many useful comments and corrections. English language has been carefully proofread. In this revising chance, we revised the manuscript over and over again, besides all revisions shown above, there are other modifications marked in bold font at “Moreno Bold” version, which are shown as below:
Lines 33-34 have been rewritten as “is of particular concern, especially among 'ESKAPE' organisms: Enterococcus faecium”.
Line 51: “production of specific proteins that help” has been rewritten as “Also, they produce specific proteins that help”.
Line 63 has been revised as “to eradicate the already formed ones, are mandatory”.
Line 69 has been revised as “molecules, to fight against MDR bacteria”.
Lines 73-74 have been revised as “take a long time to be developed [4]. Thus, to reduce the problem of antibacterial resistance in a short period, will be challenging”.
Line 76 has been revised as “but their use have declined”.
Line 82 has been rewritten as “These NPs have shown a broad”.
Line 85 has been rewritten as “NPs are able to inhibit the growth of bacteria by inhibiting”.
Lines 98 -100 have been rewritten as “Hydrogels are capable to improve cellular internalization [29], to absorb wound exudates [42], to expedite skin healing, to stimulate the collagen proliferation [40], and to exhibit antibiofilm activity”.
Line 107: “hydrogel” has been replaced by “polymer based hydrogel”.
Line 119: “the hydrogel” has been rewritten as “the hydrogels”.
Line 132: “Ag NPs can produce harm” has been rewritten as “Ag NPs can cause harm”.
Line 140: “produce free radical species” has been rewritten as “produce free radical”.
Line 152: “interaction behavior with cells” has been rewritten as “interaction with cells”.
Line 169 has been rewritten as “and the negatively charged bacterial membrane”.
Line 180: “and also release the drug” has been revised as “and release the drug”.
Line 217: “Other methodologies” has been revised as “the methods used to improve chitosan processability”.
Line 219: “by means the chemical reactions” has been revised “by chemical reactions”.
Line 249 has been revised as “Ag/chitosan-carboxymethyl β-cyclodextrin hydrogel (CM- βCD) is an alternative”.
Line 253: “The zone inhibition” has been rewritten as “The inhibition zone”.
Line 267: “is responsible of self-healing property acquired in Ag NPs” has been rewritten as “was responsible for self-healing property in Ag NPs”.
Line 274: “The pore size hydrogel” has been rewritten as “The hydrogel pore size distribution”.
Lines 306-307: “a greater amount of NPs and avoided their aggregation of them onto its surface” have been rewritten as “a higher number of NPs and avoided their aggregation onto its surface”.
Line 315: “using concentration de 0.25 mM de AgNO3” has been rewritten as “ when AgNO3 concentration was 0.25 mM”.
Line 348: “The different shapes NPs may have a” has been revised as “Different NPs shapes may present a”
Line 349: “leading to an antibacterial activity vary” has been rewritten as “leading to a diverse antibacterial activity”.
Line 462: “The hydrogel encapsulating Au NPs” has been revised as “Au NPs encapsulated hydrogels”.
Line 468: “Important characteristics Au NPs such as” has been rewritten as “Some important characteristics of Au NPs, such as”.
Lines 479-480 have been rewritten as “The incorporation of Au NPs into biocompatible supports, such as Liposomes, is one of the approaches used in the biomedicine area”.
Line 568: “with increasing the content of Au NPs” has been rewritten as “when the content of Au NPs increased”.
Line 601: “bacteria to create a bacterial receptor that is able to” has been revised as “bacteria. This creates a bacterial receptor able to”
Lines 711-712: “1 W/cm2 power intensity with 1 cm2 laser spot size and 15 s laser pulses) was” have been revised as “A set of 1 W/cm2 power intensity, with 1 cm2 laser spot size, and 15 s laser pulses was”.
Line 732: “use of Au NPs hydrogels is their association” has been revised as “used Au NPs hydrogels in association”.
Line 775: “through porins present in” has been rewritten as “through existing porins in”.
Line 812: “that” has been revised as “All these mechanisms”.
Line 818: “to the release of silver so” has been rewritten as “to release silver. So,”
Line 843: “cell division, cause” has been revised as “cell division, causing”.
Line 909: “in addition to the composition and structure of the biofilm, including pore size,” has been rewritten as “The composition and structure of the biofilm are also important, including pore size”.
Line 917: “and reducing pathogen viability by altering” has been rewritten as “The pathogen viability is reduced by altering”.
Line 971: “incorporate two metallic” has been revised as “incorporation of two metallic”.
Thus, we believe that we considered all the comments of the Reviewer #1, and we hope that our review is now adequate for publication in the MDPI Antibiotics.
Thanking you very much for the attention given to our review.
Sincerely Yours
Profa. Dra. Isabella Macário Ferro Cavalcanti
Laboratory of Microbiology and Immunology
Academic Center of Vitória (CAV)
Institute Keizo Asami (iLIKA)
Federal University of Pernambuco (UFPE), Brazil
https://orcid.org/0000-0002-7889-3502

Reviewer 2 Report
This article the Authors reviews the most recent investigations on the characteristics, applications, advantages, and limitations of hydrogels combined with metallic NPs for treating MDR bacteria. The mechanisms of action of the antibacterial and antibiofilm activity of the NPs incorporated in hydrogels were also described.
The manuscript submitted for review is very well prepared. The only thing I can point out is the lack of drawings that would definitely facilitate the reading and understanding of certain content in the presented manuscript. In my opinion, illustrative drawings should be added to the manuscript.
Author Response
Dear Reviewer,
Please see the attachment.
Prof. Dr. Sotiris K Hadjikakou
Dr. Christina N. Banti
Guest Editors of Antibiotics
Ms. Thalia Zhang
Assistant Editor, MDPI
Vitória de Santo Antão, December 21st 2022.
Dear Editors and Referees,
Thanks for your message with the report of the Referees of our review entitled Advanced Hydrogels Combined with Silver and Gold Nanoparticles Against Antimicrobial Resistance by Moreno et al., to be submitted to the special issue of MDPI Antibiotics entitled “Silver and Gold Compounds as Antibiotics”.
We have carefully taken into account all the comments and suggestions made by them. We agree with all of them and we sincerely believe that they improved the quality of our work. We explain below all the changes which were done, as well as the place in the manuscript they were inserted, according to each comment. Please check the following our reply (blue font) of each reviewer’s comments (black and italic font). For the sake of clarity, a marked “Moreno Bold” version is also included.
Reviewer #2:
This article the Authors reviews the most recent investigations on the characteristics, applications, advantages, and limitations of hydrogels combined with metallic NPs for treating MDR bacteria. The mechanisms of action of the antibacterial and antibiofilm activity of the NPs incorporated in hydrogels were also described.
The manuscript submitted for review is very well prepared. The only thing I can point out is the lack of drawings that would definitely facilitate the reading and understanding of certain content in the presented manuscript. In my opinion, illustrative drawings should be added to the manuscript.
Following the reviewer’s advice, the next figures were added:
Figure 1 has been added at line 324 on page 7.
Figure 1. Silver and gold NPs loaded into hydrogels for antibacterial application.
Figure 2 has been added at line 758 on page 14.
Figure 2. The mechanisms of bactericidal action of Au NPs and Ag NPs loaded into hydrogel.
Figure 3 has been added at line 838 on page 16.
Figure 3. Mechanism of inhibiting biofilm formation of AuNPs and AgNPs.
Figure 4 has been added at line 885 on page 17.
Figure 4. Mechanism of biofilm eradication of Au NPs and Ag NPs.
The review title has been rewritten as “Advanced Hydrogels Combined with Silver and Gold Nanoparticles Against Antimicrobial Resistance”. Furthermore, English language has been carefully proofread. In this revising chance, we revised the manuscript over and over again, besides all revisions shown above, there are other modifications marked in bold font at “Moreno Bold” version. The suggestions of all reviewers were gratefully accepted. We believe the review has been considerably improved and is now adequate for publication in MDPI Antibiotics.
Thanking you very much for the attention given to our review.
Sincerely Yours
Profa. Dra. Isabella Macário Ferro Cavalcanti
Laboratory of Microbiology and Immunology
Academic Center of Vitória (CAV)
Institute Keizo Asami (iLIKA)
Federal University of Pernambuco (UFPE), Brazil

Reviewer 3 Report
The review presents in a well-organized manner, with details, the antimicrobial effects of different systems of hydrogels combined with Ag or Au nanoparticles.
Minor comments:
The title should reflect more precisely the Ag and Au complexes (as this is the content of the review), not metals overall. And the topic does not address antibiotic resistance specifically, but antimicrobial effects overall.
It is not clear if there is a difference between hydrogels "incorporated" and "loaded" with NP.
Author Response
Dear Reviewer,
Please see the attachment.
Prof. Dr. Sotiris K Hadjikakou
Dr. Christina N. Banti
Guest Editors of Antibiotics
Ms. Thalia Zhang
Assistant Editor, MDPI
Vitória de Santo Antão, December 21st 2022.
Dear Editors and Referees,
Thanks for your message with the report of the Referees of our review entitled Advanced Hydrogels Combined with Silver and Gold Nanoparticles Against Antimicrobial Resistance by Moreno et al., to be submitted to the special issue of MDPI Antibiotics entitled “Silver and Gold Compounds as Antibiotics”.
We have carefully taken into account all the comments and suggestions made by them. We agree with all of them and we sincerely believe that they improved the quality of our work. We explain below all the changes which were done, as well as the place in the manuscript they were inserted, according to each comment. Please check the following our reply (blue font) of each reviewer’s comments (black and italic font). For the sake of clarity, a marked “Moreno Bold” version is also included.
Reviewer #3:
The review presents in a well-organized manner, with details, the antimicrobial effects of different systems of hydrogels combined with Ag or Au nanoparticles.
Minor comments:
- The title should reflect more precisely the Ag and Au complexes (as this is the content of the review), not metals overall. And the topic does not address antibiotic resistance specifically, but antimicrobial effects overall.
As suggest by Reviewer #3, the title has been rewritten as “Advanced Hydrogels Combined with Silver and Gold Nanoparticles Against Antimicrobial Resistance”.
- It is not clear if there is a difference between hydrogels "incorporated" and "loaded" with NP.
We agree the reviewer’s comment, there are difference between “incorporated” or “loaded”. Incorporated involves in situ synthesis of nanoparticles. Loaded is the addition of the nanoparticles into the hydrogel, which NPs can be previously synthesized out or also can be in situ synthesis within hydrogel. To avoid confusion in the text, some titles of the review have been revised as:
Line 196 has been rewritten as “2.1 Antibacterial activity of Ag NPs loaded in hydrogels”.
Line 379 has been rewritten as “Table 2. Ag NPs loaded into hydrogel for antibacterial application”.
Line 477 has been rewritten as “3.1 Antibacterial of Au NPs loaded in hydrogel”.
Line 686 has been rewritten as “Table 3. Au NPs incorporated into hydrogel for antibacterial application”.
Line 690 has been rewritten as “3.2 Antibiofilm activity of Au NPs loaded in hydrogel”.
Done.
We would like to thank you very much for reviewing our manuscript carefully and giving us many useful comments. Furthermore, English language has been carefully proofread. In this revising chance, we revised the manuscript over and over again, besides all revisions shown above, there are other modifications marked in bold font at “Moreno Bold” version. The suggestions of Reviewer #3 were gratefully accepted. We believe the review has been considerably improved and is now adequate for publication in MDPI Antibiotics.
Thanking you very much for the attention given to our review.
Sincerely Yours
Profa. Dra. Isabella Macário Ferro Cavalcanti
Laboratory of Microbiology and Immunology
Academic Center of Vitória (CAV)
Institute Keizo Asami (iLIKA)
Federal University of Pernambuco (UFPE), Brazil

Round 2
Reviewer 1 Report
Congratulations on the revised manuscript! I noticed that you took into account my observations!